# Seismic precursors to the Whakaari 2019 phreatic eruption are transferable to other eruptions and volcanoes

Alberto Ardid [1✉], David Dempsey [1], Corentin Caudron[2] & Shane Cronin [3]

Volcanic eruptions that occur without warning can be deadly in touristic and populated areas. Even with real-time geophysical monitoring, forecasting sudden eruptions is difficult, because their precursors are hard to recognize and can vary between volcanoes. Here, we describe a general seismic precursor signal for gas-driven eruptions, identified through correlation analysis of 18 well-recorded eruptions in New Zealand, Alaska, and Kamchatka. The precursor manifests in the displacement seismic amplitude ratio between medium (4.5–8 Hz) and high (8–16 Hz) frequency tremor bands, exhibiting a characteristic rise in the days prior to eruptions. We interpret this as formation of a hydrothermal seal that enables rapid pressurization of shallow groundwater. Applying this model to the 2019 eruption at Whakaari (New Zealand), we describe pressurization of the system in the week before the eruption, and cascading seal failure in the 16 h prior to the explosion. Real-time monitoring for this precursor may improve short-term eruption warning systems at certain volcanoes.

[1] University of Canterbury, Christchurch, New Zealand. [2] Université Libre de Bruxelles, Bruxelles, Belgium. [3] University of Auckland, Auckland, New Zealand.
✉email: aardids@gmail.com

Gas-driven explosions in volcanic areas usually occur with little warning and have recently caused loss of life in Japan (2014 Mt Ontake eruption, 60 deaths[1]) and New Zealand (2019 Whakaari eruption, 22 deaths[2]). Whakaari is an andesitic stratovolcano island located at the northern end of New Zealand's Taupo Volcanic Zone (TVZ) (Fig. 1a). It is one of the country's most active volcanoes with a long history of fumarolic activity, interspersed with phreatic, phreatomagmatic, and magmatic eruptions[3,4]. Because of this history, and despite real-time monitoring, identification and early warning of eruption precursors at this and similar volcanoes remains challenging.

Phreatic eruptions are usually driven by expansion and steam-flashing of suddenly released hot pressurized fluids[5]. Containment arises through sealing of a liquid water, magmatic gas, or steam reservoir under diverse conditions, e.g., low-permeability layers that form below crater lakes or within crater basins (clay/silt deposits, liquid sulfur[6]); hydrothermal alteration that changes rock texture/permeability[7]; pore-blockage by elemental sulfur or hydrothermal precipitates[8–10]; and/or active sealing by pressure applied to compressible clays[11]. Permeability barriers are especially important if gas input increases during magmatic unrest. Restriction of gas flow through the upper vent system leads to localized shallow pressurization, creating the conditions for an explosive eruption.

Hydrothermal seals and their failure can play an important role in phreatic eruptions because diminished vent permeability drives pressurization whereas a rapid increase can drive sudden decompression[6,12,13]. The timescales of seal formation depend on a range of mineralization, pressurization, and fluid-rock properties[14]. In active hydrothermal systems beneath lakes, seals commonly form over weeks to months[15] with mineral precipitation (sulfur, sulfates, and silica). In extreme cases, they may form in seconds to hours[16] due to mineralized zones (alunite, anhydrite, smectite) that open and close in response to pressure variations.

Seismic data analysis is standard practice at volcano observatories to track magmatic processes, the volcano state, and the possibility of future eruptions[17–21]. Continuous volcanic seismic signals are interrogated to understand fluid movement (e.g., magma degassing), conduit fracture/bubble processes that precede eruptions, and pressurization of a hydrothermal system[22–24]. However, for continuously active, seismically noisy, and "wet" volcanoes[25], tremor precursors prior to phreatic or phreatomagmatic eruptions are difficult to distinguish, or only identified in retrospect[2,6,8,26,27]. Detection of tremor prior to Whakaari eruptions[28] has helped support diagnoses of volcanic unrest but has not been used to establish eruption imminence. This is partly due to the very shallow origin of this signals, with pressurization and triggering occurring within a few tens to hundreds of meters of the surface[5,21,29]. The signals can also be dampened when low-permeability seals form[8,30] because surface fluid flux is diminished. Such quietening of the system has the potential to impart a false sense of safety when actually the trapped fluids may be pressurizing. A sudden pulse of additional magmatic gas may rupture these systems, but eruption initiation can also be any process that causes decompression, such as surface-propagating drying/cooling cracks, tectonic fracturing, sudden overburden loss due to landslides[5] or lake breakouts.

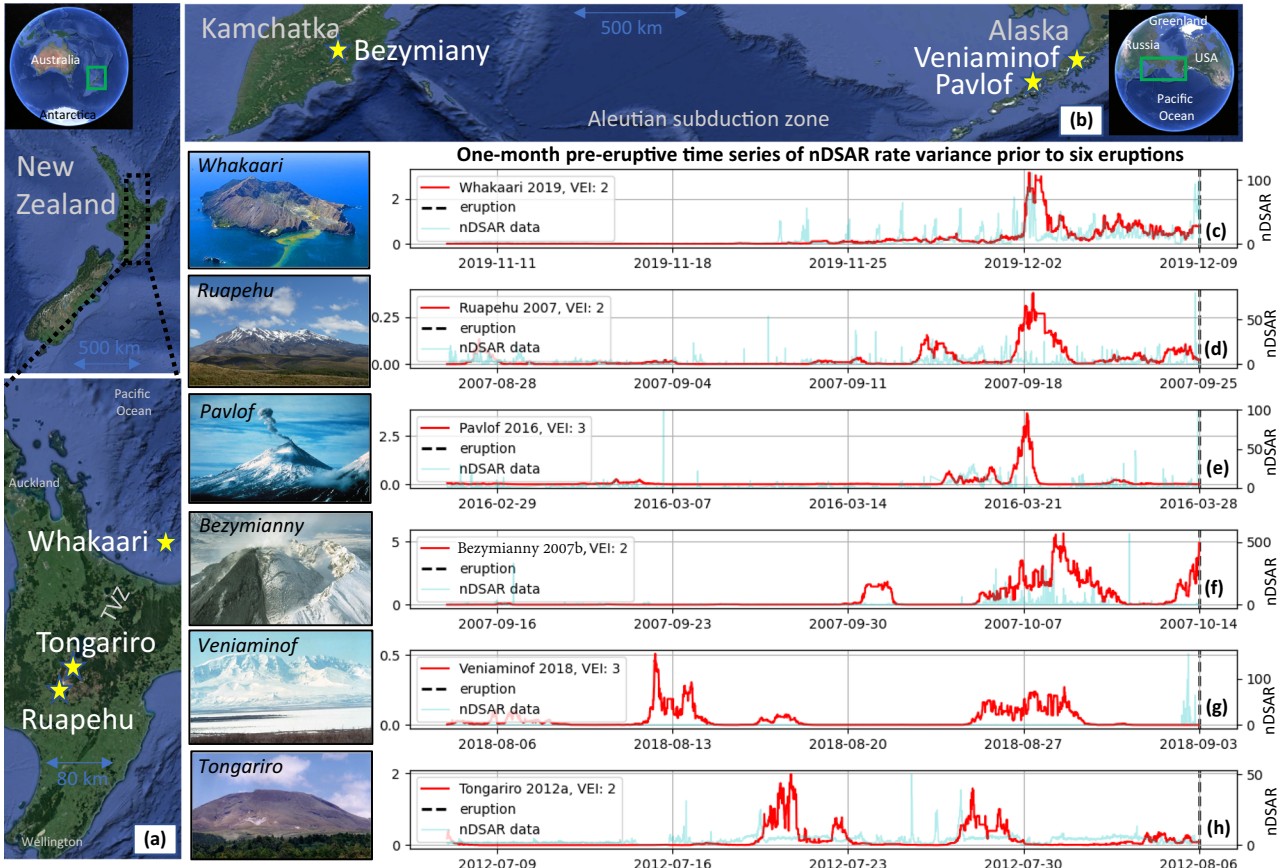

**Fig. 1 Locations of the six volcanoes in this study. a** Whakaari, Tongariro, and Ruapehu in New Zealand (TVZ: Taupo Volcanic Zone); **b** Veniaminof and Pavlof in Alaska, USA, and Bezymianny in the Kamchatka Peninsula, Russia. **c–h** Time-series for the feature nDSAR (normalized displacement seismic amplitude) rate variance (red) prior to six major eruptions (black dashed line) at each volcano (LHS y-axis indicates the feature magnitudes). The raw nDSAR data time series is shown in the background (light blue) (RHS y-axis indicates their magnitudes). VEI = Volcano Explosivity Index.

Tremor data in the frequency range 0.1–25 Hz[18,31] are often analyzed within frequency bands. Real-time seismic amplitude measurement (RSAM) and reduced displacement use the 2–5 Hz band[32] (also, wider bands[33]) for short-term forecasting[2,34–36]. Medium (MF; 4–8 Hz) and high-frequency bands (HF; 8–16 Hz), and their ratios are used to interpret seismic attenuation in relation to source effects and source changes[24]. The displacement seismic amplitude ratio (DSAR; MF/HF) has been linked to month- and year-long variability prior to eruptions[21] that is interpreted as progressive sealing and tendency to pressurization. Rather than using earthquakes, DSAR has been used to investigate changes in the persistent shallow seismic vibrations by adopting this simple band-ratio approach[24]. To evaluate these continuous signals, pattern-recognition, clustering, neural network and failure-forecasting tools are used[2,37,38]. In particular, time-series feature engineering may reveal hidden, but statistically relevant patterns in complex time series[2].

Here, we analyzed tremor time-series prior to 18 eruptions at six volcanoes: Whakaari, Ruapehu and Tongariro (New Zealand), Veniaminof and Pavlof (Alaska, USA), and Bezymianny (Russia) (Fig. 1a, b). Individual eruptions are referred to with the shorthand volcano year, e.g., Whakaari 2013a refers to the first eruption in 2013 at Whakaari. We used feature engineering to extract latent patterns from the tremor and then systematically mined these for statistically differentiable correlations across all volcanoes in the weeks prior to eruptions (see "Methods"). We identified characteristic peaks in the 2-day median of normalized DSAR tremor (nDSAR median; see Methods; n denotes normalization) that recurred prior to eruptions at several volcanoes. Corroborated by other observations, we present a timeline of the 2019 Whakaari eruption, explaining how rapid sealing in the shallow hydrothermal system promoted pressurization and created the conditions for a phreatic explosion.

## Results and discussion
**Recurrency of pre-eruptive time-series features**. In the four weeks leading up to an eruption, we observe correlated patterns across at least three classes of derived tremor time series. These patterns are the result of an initial screening using a cross-correlation analysis of eruptive periods (see "Methods"). Later, we discard spurious patterns that are not differentiable from periods of repose. Prior to six major eruptions (VEI > 2) at each of the studied volcanoes, there is a sustained elevation of nDSAR rate variance between 5 and 10 days prior to the event (Fig. 1c–h; see Fig. S1 for equivalent time series on all eruptions). This feature quantifies variance in the "spikiness" of nDSAR, however, its pattern is not obvious from visual inspection of the raw data.

A second pattern was identified in the nDSAR median one week before eruptions (Fig. 2). Patterns between Whakaari eruptions and others as Veniaminof 2013 are especially similar. Both records exemplify a steady rise of nDSAR median to a peak a few days before the eruption. The eruption itself often occurs after a day or two of decline. We quantify similarity in the pattern shape between pairs of eruptions by calculating the cross-correlation coefficient, CC, for the 4-week records prior to eruption. For instance, a CC = 0.71 was found between the Whakaari 2019 and Veniaminof 2013 eruption pair. The smoothed nDSAR pattern is broadly similar across all studied volcanoes with some variability in the timing, width, and magnitude of the peak, which produces a range of CC values (Fig. 2a). The greatest similarity is seen between Whakaari, Veniaminof, and Ruapehu volcanoes, whereas Tongariro, Pavlof, and Bezymianny, show a self-similar pattern of several decreasing cycles of the nDSAR median in the month before their eruptions.

The third pattern is observed as an increased strength of 75 min oscillations in nHF tremor, with similar timings prior to several eruptions as Whakaari 2016, 2019, Ruapehu 2009, and Bezymianny 2007a, b (see Fig. S4). These eruptions were dominated by strong activity around two days prior with inverse RSAM exhibiting a linear decline (Fig. S5), indicative of cascading material failure[34,39]. In the next section, we show that the correlation in this particular time series is spurious.

**Differentiability of eruption precursors**. Demonstrating that a common pattern occurs prior to multiple events (recurrency) is only the first step in establishing an eruption precursor. It is also transferable if it occurs prior to eruptions at other volcanoes. Qualitative inspection of nDSAR rate variance and nDSAR median suggests both are recurrent and transferable (Figs. 1 and 2). A third property of a precursor is that it should be rare (or absent) during non-eruptive unrest or volcanic repose, which we refer to here as differentiability. The statistical tests we present test the differentiability of three candidate precursors that passed the initial screening for recurrency and transferability.

First, we checked if a candidate precursor for one eruption (the archetype) had high correlation values when compared against random four-week periods of repose at other volcanoes (Fig. 3; Pavlof and Bezymianny were excluded because of their shorter data records, see Table S1). For illustration, we have used feature time-series prior to Whakaari 2019 (Fig. 2b) as archetypes, although later we show that differentiability of the precursor is independent of this choice. Figure 3a shows that the nDSAR median archetype has very high correlation values for the other four Whakaari eruptions, all exceeding the distribution 90th percentile when compared to correlation over more than 40 years of non-eruptive data. We calculated a two-sample Kolmogorov–Smirnov (K–S) p-value to check whether CC values from eruptions belonged to a similar distribution as the inter-eruptive data. The resulting value of 0.00015 is very low and suggests the eruptions come from distinct distributions. Eruptions at Ruapehu, Tongariro, and Veniaminof also rank highly in a percentile-sense, but are less differentiable than Whakaari (p-value of 0.017, 0.066, and 0.072, respectively).

A similar analysis of nDSAR rate variance suggests some differentiability of this pattern amongst the Whakaari eruptions (Fig. 3b), with all ranking above the 80th percentile (p-value of 0.006). However, the pattern is less transferable to Ruapehu, Tongariro, and Veniaminof eruptions. The 75 min nHF harmonic prior to the Whakaari 2019 eruption (Fig. S4) shows very little differentiability compared with non-eruptive data (Fig. 3c). Although strong activity is observed several days before six different eruptions (Fig. S4), this appears also to occur frequently during repose.

**Transferability of eruption precursors**. The analysis above and in Fig. 3a–c is specific to Whakaari 2019. Here, we generalize to a multi-eruption test where cross-validation is used to establish whether a precursor has differentiability independent of the archetypal eruption. For the features of interest, nDSAR median and nDSAR rate variance, we assembled different subsets of eruptions, randomly selected an archetype eruption from the subset to correlate across volcanic record, and then computed a K–S p-value to quantify its differentiability. We repeated these steps many times, each time selecting a different eruption from the subset as the archetype and applying a random "jitter" (displacing the precursors time-series backward up to 7 days; similar results were obtained with 5–8 days) to allow for time offsets. The result is a distribution of p-values that provide a general view of how differentiable a particular precursor amongst that eruption

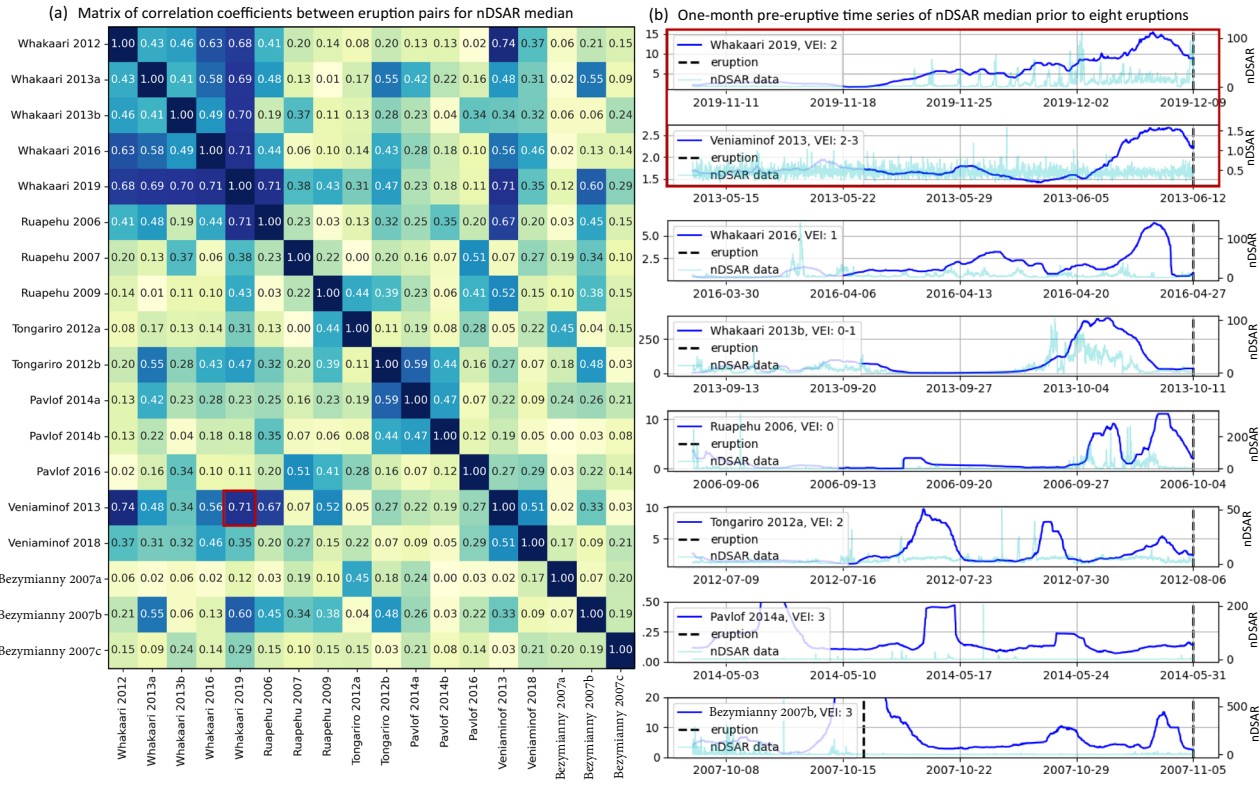

**Fig. 2 Correlation coefficients between eruption pairs for feature nDSAR median. a** Matrix of correlation coefficients between eruption pairs for the four-week nDSAR median. **b** One-month pre-eruptive time series of nDSAR median (blue) prior to eight eruptions (black dashed line; for all eruptions see Fig. S2). Raw nDSAR data are shown in light blue. The red outlined square highlights to correlation between the pair of eruptions Whakaari 2019 and Veniaminof 2013.

subset. Finally, this distribution is compared against a benchmark that was constructed by randomly selecting archetypes from within the non-eruptive data.

First, we considered the differentiability of pre-eruptive nDSAR median amongst the subset of five Whakaari eruptions. Figure 3d shows that p-values for differentiability are predominantly very small compared against a benchmark of non-eruptive data, which are more uniformly distributed up to 1. Similar results are obtained for the nDSAR rate variance. This indicates both features are robust eruption precursors for phreatic eruptions at Whakaari.

Next, we considered whether these precursors were transferable to other volcanoes. We considered the subset of phreatic eruptions at Whakaari, Ruapehu and Tongariro volcanoes as well as a mixed style subset that pools phreatic Whakaari eruptions with magmatic eruptions from Veniaminof, Pavlof, and Bezymianny. The same multi testing approach was applied.

Precursors amongst the Whakaari–Ruapehu–Tongariro set indicate a degree of differentiability (Fig. 3e), although it is less than for the subset of only Whakaari eruptions (Fig. 3d). This indicates a moderate level of transferability of the nDSAR median and nDSAR rate variance precursors between phreatic eruptions at these three volcanoes.

Precursor differentiability amongst the mixed eruption-style, Whakaari–Veniaminof–Pavlof–Bezymianny subset is much less conclusive than other subsets. Distributions of p-values for both nDSAR median and nDSAR rate variance are not obviously different from the benchmark. Clearly, there is a limit on the transferability of the eruption precursors, and this appears to be linked with eruption style. This would be expected if precursors are informative of physical changes in the volcano linked to the particular type of eruption occurring there. These physical changes are discussed in the next section.

**Inferring eruptive processes in the Whakaari 2019 eruption.** Statistical analysis supports a recurrent, differentiable, and partly transferable archetypal pattern of nDSAR median across several volcanoes (Fig. 2b), increasing to a peak ~2–4 days before an eruption. We use observations of the Whakaari 2019 eruption to explain this pattern in the context of the active hydrothermal system and aquifer within crater-filled deposits above the vent. Specifically, we identify transitions between five phases prior to the eruption: (1) interaction and gas exchange between magmatic and geothermal systems, (2) pulsating gas release at the surface, (3) consolidation of a seal, (4) aquifer pressurization, and (5) seal breakdown and eruption (Fig. 4; see Table S3).

A sustained RSAM harmonic signal lasting minutes to days (identified as "volcanic tremor"[17]), was detected at Whakaari between Nov 10 and 23 (phase 1, Fig. 4c). This signal is thought to indicate interactions between the magma source (estimated to be 0.8–1.0 km deep at Whakaari[29]) and the groundwater (geothermal aquifer). At the surface[28] geysering began and the crater lake level rose. On Nov 18, the Volcano Alert Level (VAL) was raised from 1 to 2, its highest non-eruptive category[40].

Transition to a new phase is marked by an RSAM decrease, and concurrent increases in MF and HF after Nov 23. This was accompanied at the surface by more frequent gas emissions, water jetting and a stabilization of the lake level[41], along with elevated $SO_2$ flux detected from satellite observations[42]. Short, day-long cyclic oscillations are evident in RSAM, MF, and HF bands (Fig. 4c). Since the MF signal is stronger, the oscillations are associated with short-term rises in the unsmoothed DSAR (Fig. S7). Each cycle ends with a linear decline of inverse RSAM (Fig. 4a), which suggests that cascading material failure is occurring[39].

Taken together, these observations imply that the Nov 23 to Dec 02 period was dominated by pulsatory gas fluxing. Each cycle

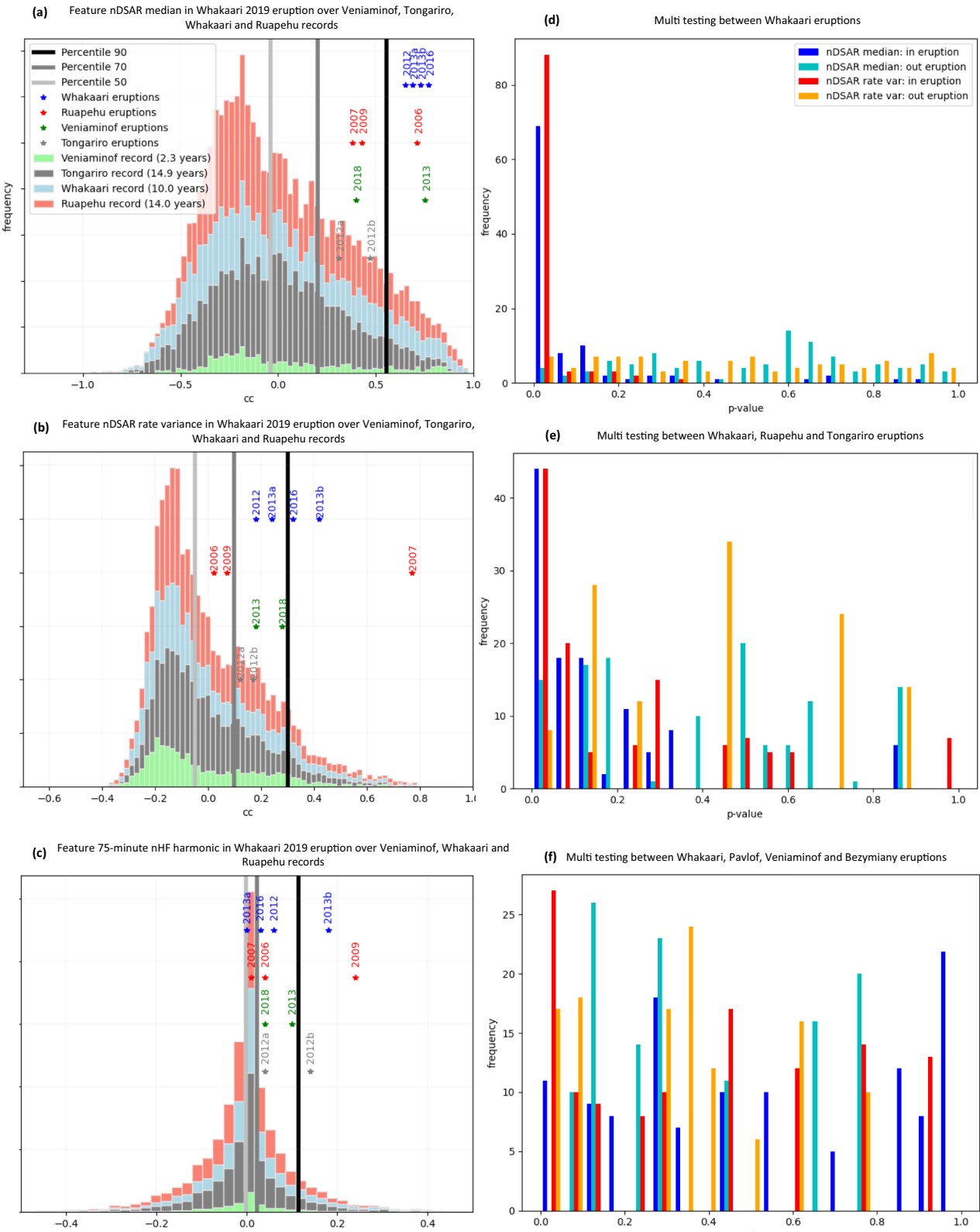

begins with building MF and DSAR which likely represents pressurization below a blocked conduit or weak seal. The cycle ends with cascading seal failure that renews gas flow paths, and generates surface jetting, accompanied by decrease in MF and DSAR. Sealing in the breccia-filled crater and conduit could be due to deformable clays, pore blockage by elemental sulfur, or rapid sulfate mineral precipitation along thin cracks. Seal failure

and crack formation at Whakaari is likely transient[16]. On Dec 02 there was an especially large gas pulse as seen in $SO_2$ flux[42], which coincides with a strong signal in the smoothed nDSAR rate variance (Fig. 4a). This marked the end of the pulsatory phase and may be indicating a change in the system state to be more conducive to voluminous sealing or pressurization with comparatively larger gas explosions.

**Fig. 3 Testing differentiability of potential precursors. a–c** Percentile analysis of correlation coefficients (cc) for Whakaari 2019 archetype features nDSAR median, nDSAR rate variance, and nHF harmonic 75 min, respectively. The distribution of daily correlation coefficients calculated using these archetypes are shown over the available Whakaari (blue), Ruapehu (red), Tongariro (gray), and Veniaminof records (green). Correlations with other eruptions are labeled by year of eruption and corresponding volcano color. Vertical offset is for distinction only. Median, 70th and 90th percentiles are shown as increasingly darker gray vertical bars. **d–f** Statistical multi-testing and cross-validation of archetypes across different eruption pools. **d** Five Whakaari eruptions. **e** Phreatic eruptions from Whakaari, Ruapehu, and Tongariro. **f** Phreatic Whakaari and magmatic Veniaminof, Pavlof, and Bezymianny eruptions. Distribution of K–S $p$-values from repeat testing of nDSAR median and nDSAR rate variance archetypes prior to eruptions (denoted "in eruption"; blue and red) and randomly selected from the non-eruptive record (denoted "out eruption"; cyan and yellow). Increasing differentiability is indicated by a distribution that clusters closer to zero.

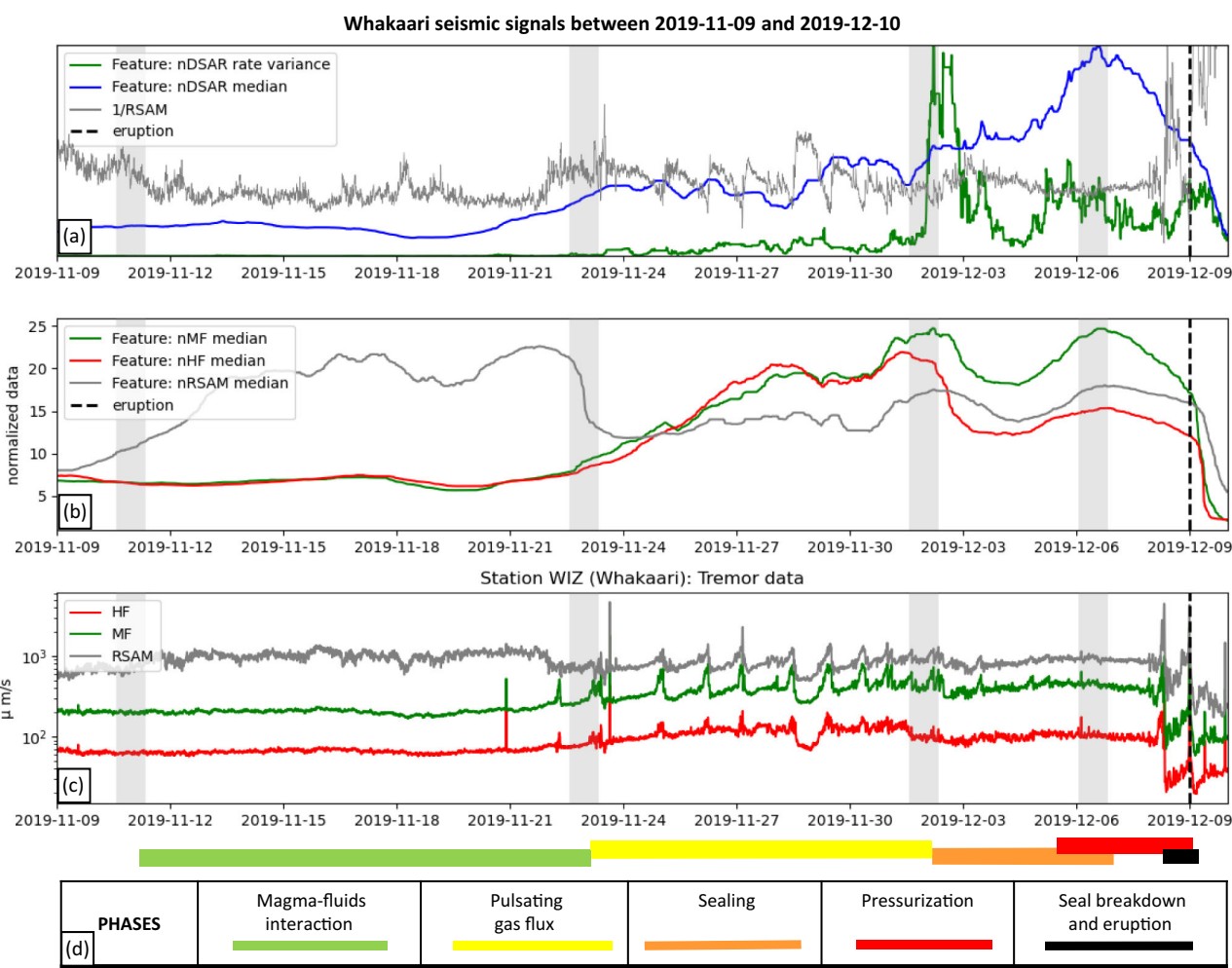

**Fig. 4 Four-week pre-eruptive time series for the Whakaari 2019 eruption. a** nDSAR median (blue), nDSAR rate variance (green), and inverse RSAM (gray); **b** 2-day moving medians for normalized MF (green), HF (red), and RSAM (gray); **c** raw, unsmoothed MF, HF, and RSAM. Vertical gray bands mark approximate divisions of the five phases labeled at the base of the figure. **d** Identified transitions between five phases prior to the eruption.

After Dec 03, both the nMF and nHF medians (2-day normalized smoothed MF and HF tremor) decline for a few days, before a reversal and steady rise of the MF signal (phase 3, Fig. 4b). Decoupling of MF and HF signals during this period is reflected in a concurrent increase of the nDSAR median (Fig. 4a), identified above as a key eruption precursor at Whakaari. There was no SO₂ flux[42] or elevated surface activity reported during this period and we hypothesize that this reflects more efficient seal consolidation and pressurization. This is consistent with prior association between short-term MF increase and pressurization in the pulsatory phase, but now with a larger disparity between MF and HF. A sub-surface seal will suppress fluid release, and in-turn dampen surface processes (decreases HF signal). Ongoing fluid entry and pressurization below the seal causes stronger MF

return, because HF is attenuated as signal transits from the subsurface. The mechanism of seal consolidation is uncertain but could be, e.g., determined by analysis of ballistics[43].

The fourth phase is characterized by a reversal of the nDSAR median, which declines steadily in the days before eruption (phase 4, Fig. 4a). The decline mainly reflects a proportionally larger drop in MF (Fig. 4b). This 'MF-quieting' could indicate that the reservoir/aquifer below the seal has reached an equilibrium pressure with the current state of deep fluid recharge.

On Dec 08, ~16 h before the eruption, a burst of activity occurred across all tremor bands (Fig. 4c, waveforms in Fig. S8). At the same time, inverse RSAM began a characteristic linear decrease (Fig. 4a). This pattern has been used for retrospective[34] forecasting of Whakaari eruptions using the failure forecast

method[39,44]. As with the pulsatory phase, this decline is interpreted as cascading material failure, including crack formation within the seal; this eventually allows explosive escape of the pressurized fluids. Notable is a strong $SO_2$ signal up to 40 min before the eruption[42], possibly reflecting the final stages of seal breakdown. The seismic activity burst 16 h before eruption was recognized[2] as a precursor and hypothesized to represent a sudden gas/fluid influx. We show here, however, that pressurization and pre-conditioning of the system to eruption can be recognized much earlier.

**Final comments**. Shallow-seated sudden explosive eruptions are notoriously difficult to forecast, especially in wet and noisy volcanoes. We performed a correlation analysis across feature time-series derived from tremor data in the weeks preceding 18 explosive eruptions across six volcanoes. We showed that there are common feature patterns, two of which can be classed as phreatic eruption precursors because they are recurrent and differentiable from non-eruptive repose. The nDSAR median feature is potentially also transferable to other volcanoes.

Both nDSAR median and nDSAR rate variance features are especially similar across five Whakaari eruptions, and also recognized prior to Ruapehu, and Tongariro eruptions. We used the Whakaari 2019 eruption to relate these precursors with hydrothermal reservoir sealing and pressurization that prime the system for a phreatic eruption. Five distinct phases track the onset to such shallow gas and steam-driven eruptions: (1) deep source fluid input into a geothermal reservoir; (2) pulsatory gas fluxing with weak temporary sealing over one or more weeks; (3) strong seal formation over the hydrothermal system several days before the event (4) leading to a critically pressured shallow aquifer (in near-equilibrium with deep gas pressures); (5) external perturbation, and/or cascading material failure and crack formation over ~10–20 h, leading to final seal breach and explosive decompression.

Our retrospective analysis suggests these precursors have encouraging properties for short term eruption forecasting. This could include real-time monitoring for individual precursors, or using models that optimally combine them. However, true forecasting skills must be demonstrated with prospective experiments, which underscores how operationalization, testing, and post hoc analysis are essential complements to predictive volcanology. This is the best way to produce trusted forecasts that can improve safety at inhabited or regularly visited volcanoes.

## Methods

**Tremor data processing**. Continuous seismic waveform data were downloaded using ObsPy[45] for six broadband stations close to the craters or vents of subject volcanoes (see Table S1). In total, 50 years of data were processed, ensuring that each eruption had at least one month of data prior (Table S2).

The instrument response was removed from the daily waveform data and this was processed to obtain four time-series (data streams): RSAM (real-time seismic amplitude measure), a 10 min average of the absolute vertical station velocity bandpass filtered between 2 and 5 Hz; MF (medium frequency) and HF (high-frequency), computed the same as RSAM but using the 4.5–8, and 8–16 Hz frequency bands; and DSAR (displacement seismic amplitude ratio), the ratio between MF and HF bands recalculated for absolute vertical displacement. DSAR is computed by integrating the MF and HF signals, then computing the average absolute signals over 10 min windows, and then taking the ratio of the two quantities[2,24]. The 4–5 Hz threshold between RSAM and MF reflects an assumption that tremor mostly radiates energy below 4.5 Hz. To exclude this effect and explore attenuation related to permeability change (such as sealing), this frequency value is used as a threshold.

To isolate continuous tremor signal, outlier detection was used to remove regional and volcano-tectonic (VT) earthquakes before each of the above data streams were calculated. Wave trains from earthquakes perturb the average velocity record at stations above the background. After averaging, there is an outlier in the RSAM that adds noise and confuses algorithms without context. To identify and remove the earthquake effects, we applied an outlier detection algorithm to raw velocity traces, excising a 2 min envelope whenever velocity exceeded three

standard deviations from the data 10 min window mean. This removed most short duration events but still resulted in contamination by larger subduction earthquakes. To remove these, we applied a two-window (20 min), moving-minimum. Other methods to automatically identify earthquake signals[46] could also be used.

For comparison between volcanoes, values in each data stream are transformed to a unit log-normal distribution (or a unit normal distribution in $log_{10}$-space). This is called z-score normalization and it has been shown to improve classification modeling[47]. Thus, a normalized value of 1 indicates a datapoint at the mean, and a value of 100 indicates the signal is two standard deviations above the mean. Alternative normalization techniques based on station distance could be implemented[32,48].

**Feature extraction**. Each of the four data streams was subdivided into overlapping 48 h windows, with each window advancing one data point (10 min) beyond the previous one. For each window, time-series features are extracted using the Python package tsfresh[49]. Features included measures of distribution (parametric: mean, standard deviation; and non-parametric: quantiles, number of peaks), auto-correlation, frequency content, linearity, and information content (entropy, energy, nonlinearity scores). Feature extraction yields 524 new time-series for each data stream. This is fewer than the 700 features[2] because we excluded some linear regression features with high correlation, FFT coefficients higher than the Nyquist frequencies, and FFT phase angles. The effect of window length was explored[2] and shown not to affect the resolution of short-term eruption precursors at Whakaari.

The feature nDSAR median corresponds to a time series calculated as the 'median' from the normalized DSAR data in a 48 h window. The feature referred to as nDSAR rate variance is labeled in the tsfresh Python library as 'change_quantiles__f_agg_"var"__isabs_False__qh_0.6__ql_0.4'. It is calculated as the variance of adjacent data differences (the "rate") of nDSAR values falling in the interquartile range 0.4–0.6 (hence, outliers are removed). The feature referred to as 75 min nHF harmonic has the corresponding tsfresh label 'fft_coefficient__coeff_38__attr_"real"' and is the real component of Fourier coefficient 38, which corresponds to a frequency with period of approximately 75 min. See 'Quick recipes' in the Supplemental Material for a straightforward procedure to calculate the DSAR median and the DSAR rate variance.

The tsfresh library calculates many features. However, there are 'families' of features that capture similar characteristics and are highly correlated with one another. For example, the 'median' feature (i.e., Fig. 2) has strong similarities with other time series quantiles. When undertaking eruption correlation analysis, we obtained quite similar results for DSAR quantiles 0.3, 0.4, 0.5 (median), 0.6, and 0.7. We decided to use the median (0.5 quantile) as it is more understandable. A similar situation occurs for the DSAR rate variance (that measures variance of adjacent data differences falling in the interquartile range 0.4–0.6), which has quite similar patterns across other quantile ranges (i.e., ~0.3–0.7).

**Cross-correlation analysis**. For each feature time-series, e.g., nDSAR median, we extracted the four-week period prior to each of the 18 eruptions in this study (Table S2). We calculated the standard correlation coefficient (Pearson method) between each pair of pre-eruption feature data, obtaining a matrix (see Fig. 2a as reference). We also calculated Kendall and Spearman correlations, obtaining similar results (Fig. S12; notice the correlation matrices are only a first guide for selecting the features to be later tested as precursors). Once matrices were assembled for all features (~2000), we ranked them by highest average correlation, grouping similar features together (e.g., median, mean, and sum). We selected from this ranking to create a shortlist of features for visual inspection (Figs. 1 and 2b) and the next stage of testing.

The correlation method is sensitive to temporal offsets of signals or stretching on the time axis. The former is partially compensated by introducing a "jitter" at multi-testing. However, the method is only used for the initial screening of potential precursors, and spurious correlations are identified at the next stage of statistical testing. Sophisticated alternatives such as maximum convolution or dynamic time-warping may reveal other patterns missed by our approach.

**Statistical tests**. We conduct statistical analysis to test whether a pre-eruptive feature that correlates across multiple eruptions is also differentiable from non-eruptive repose periods. This verifies whether a result generated by testing the data is not likely to occur randomly, but is instead attributable to a specific cause.

The first test is the differentiability of a specific precursor archetype when comparing it to other eruptions as well as the non-eruptive data. First, we extract a feature time-series (the archetype) for the four-weeks prior to a nominated eruption. The archetype is used to calculate a correlation coefficient with four-week periods of the same feature over the entire record, shifting forward one day at a time to calculate a new correlation coefficient (this calculation is performed most efficiently as a convolution). For example, approximately 10 years of Whakaari data yields about 3600 correlation values for each feature. Histograms are used to summarize the overall distribution and percentiles to identify where individual eruptions fall on the distribution (Fig. 3a–c). High percentile values for eruptions are an indication of dissimilarity from the background distribution. This is quantified by a two-sample K–S p-values, with correlation coefficients from

eruptions in one sample and non-eruptive periods in the other. We tested different archetype lengths (three and five weeks) and found that the results were similar.

As the selection of a particular eruption as the archetype is arbitrary, we repeated the process using a different eruption each time, a form of cross-validation. To allow for uncertainty in the exact eruption time or for precursors to have time offsets (e.g., inspection feature time series in Figs. 1 and 2 indicates the timing of peaks can vary by a few days), we applied a "jitter" to each archetype. This was a random displacement backwards by up to 7 days (we tested values between 5 and 10 days and saw no change in results). The K–S p-value was then calculated for the new archetype in the manner described in the previous paragraph. Eruptions are pooled from different volcano subsets, with each eruption taking a turn as the archetype, and the same eruption tested multiple times with different jitters. The result is a distribution of p-values that represents all eruptions in the subset and allows for time offsets.

As a benchmark for comparison, a second K–S p-values distribution is constructed except that, instead of selecting the archetype as a four-week feature time-series before an eruption, a random four-week feature is chosen from the entire volcano record. Archetypes chosen this way will have low differentiability by construction. Thus, the benchmark adds a degree of robustness for interpreting differentiability, because it outlines what a non-differentiable eruption precursor looks like.

The following eruption subsets were tested for differentiability:

1. All phreatic eruptions at Whakaari for two features of interest, nDSAR median, and nDSAR rate variance. Each of the five eruptions is correlated with the Whakaari record and a K–S p-value calculated. We iterate these 10 times, each time applying a random set-back of the archetype, for a total of fifty tests. This is compared against fifty tests with randomly selected archetypes.

2. All phreatic eruptions from Whakaari, Ruapehu, and Tongariro volcanoes. To balance the eruptions subset, a set of six eruptions is constructed, two each randomly selected from the three volcanoes. Within each set, each eruption takes a turn as the archetype and then a K–S p-value calculated after convolution over all three volcano's records. This is iterated 10 times, using a different set of six eruptions each time.

3. A mixed set of five phreatic Whakaari eruptions and eight magmatic eruptions from Veniaminof, Pavlof, and Bezymianny. Balanced subsets are constructed by sampling two eruptions from each volcano and then proceeding the same as subset 2 above.

The two-sample K–S test presumes samples contain independent observations. In practice, correlation coefficients computed on non-eruptive windows can have significant overlap. We checked if this was affecting our results by recalculating the tests using fewer, non-overlapping windows for non-eruptive data (Fig. S13). The results are not materially different to Fig. 3.

## Data availability

Raw waveform data for Ruapehu, Tongariro, and Whakaari volcanoes can be download from GEONET (https://service.geonet.org.nz and for Veniaminof, Pavlof, and Bezymianny volcanoes from IRIS (http://service.iris.edu). Both are operable as clients though the FDSN webservice (https://www.fdsn.org/networks/). Rainfall data are available in New Zealand's National Climate Database (https://cliflo.niwa.co.nz/).

## Code availability

The codes required to replicate the results of this study are available at https://github.com/aardid/volc_forecast_tl/tree/transfer-learning, released under the MIT license.

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

## Acknowledgements

Tremor data were supplied by the GEONET (New Zealand) and AVO (Alaskan Volcano Observatory, USA) seismic monitoring system. Photos from Fig. 1 correspond to: (c) Whakaari (De James Shook, CC BY 2.5, https://commons.wikimedia.org/w/index.php?curid=8645905), (d) Ruapehu (De Michal Klajban - Trabajo propio, CC BY-SA 4.0, https://commons.wikimedia.org/w/index.php?curid=39988630), (e) Pavlof (By U.S. Fish and Wildlife Service - images.fws.gov, Public Domain, https://commons.wikimedia.org/w/index.php?curid=3534132), (f) Bezymianny (De Dan Miller, U.S. Geological Survey -

http://www.volcano.si.edu/world/volcano.cfm?vnum=1000-25=&volpage=photos&photo=024049, Public Domain, https://commons.wikimedia.org/w/index.php?curid=3917792), (g) Veniaminof (De Hood R E, U.S. Fish and Wildlife Service - http://www.public-domain-image.com/public-domain-images-pictures-free-stock-photos/nature-landscapes-public-domain-images-pictures/mountain-public-domain-images-pictures/mountain-veniaminof-black-lake-landscape.jpg, Public Domain, https://commons.wikimedia.org/w/index.php?curid=24911753), (h) Tongariro (De The original uploader was Grutness de Wikipedia – Commons., CC BY-SA 3.0, https://commons.wikimedia.org/w/index.php?curid=3209038). (a, b) Map data from ©2021 Google (modified from Google Earth).

## Author contributions

A.A. and D.D. developed the time series feature engineering, designed and implemented the correlation and the statistical tests. C.C and S.C. contribute to the analysis and interpretation of time series features and their applications to the Whakaari 2019 eruption. All authors contributed to manuscript development.

## Competing interests

The authors declare no competing interests.
