## [Peer Review File · Nature Communications]

REVIEWER COMMENTS

Reviewer #1 (Remarks to the Author):

Review of the paper

Seismic Precursors to the Whakaari 2019 Phreatic Eruption detected in Five Other Volcanoes by Alberto Ardid, David Dempsey, Corentin Caudron, and Shane Cronin

The paper addresses an important scientific and societal issue, i.e., the forecast of gas-driven small eruptions that can cause casualties due to the difficulty in finding good precursors (sometimes these events are named "blue-sky" eruptions).

Although I am overall positive about the goal and the work made by the authors (the paper is well written and clear), I think that the conclusions are weakly supported by the analyses. I hope that this review may help the authors in their work.

My main concern is about the statistical testing phase which seems to be poorly described, rather weak, and maybe wrong in some aspects.

Below I summarize below my main concerns.

- **CALCULATING THE PEARSON CORRELATION COEFFICIENT.** The use of the Pearson correlation may induce spurious correlations when correlating time series that do not have a white spectrum (e.g., if they have a trend or if they are autocorrelated). In practice, we can find a high correlation between any time series with increasing trend no matter if the trends are due to completely different causes. In this case, the correlation coefficient does not represent a similarity between two time series. Moreover, the Pearson correlation is a parametric measure, and it is not clear its meaning when the data are not distributed according to a normal distribution. These are not necessarily problems, but it could be good to check the autocorrelation of each dataset. If this is not a white noise, it is important to explain why the trend in the dataset has to be taken into consideration in calculating the correlation. Moreover, the use of a nonparametric correlation coefficient like Kendall (or Spearman).

- **THE SIGNIFICANCE LEVEL OF THE TEST.** This is a very important part of the paper, but I think that it is not properly treated by the authors.

* First, if the authors do many tests, the significance level of the test has to be modified accordingly taking into account of the multiple tests. In other words, the significance level of the test makes sense if we perform the test once. If we repeat the test with some variations, or using different eruptions, the significance level of the test is not anymore the one that has been declared. I do not deepen further this issue that is very well described in statistical literature.

* Second, the authors use for the tests the patterns that have the highest correlations with other eruptions. This choice can be done if we aim to find the "common patterns", but it makes very complicated to test the statistical significance, because we are arbitrarily making a cherry picking of the patterns that have to be tested. This problem is pervasive in many fields, and it is the main motivation because models/patterns are estimated in a subset of data and tested (once) in another independent subset. This is not what the authors did (the authors compare the pattern of one single eruption with the others, leading unavoidably to the multitesting problem discussed before); and the data for the testing have also been used to find the precursory patterns.

* Third, the p-value is wrongly introduced and interpreted. There is a wide literature on the "p-value hacking", which is based on the wrong interpretation of the p-value. In essence, the p-value is not the "significance of the test", because the p-value is the probability to observe the data given the null hypothesis, whereas the second is the probability of the null hypothesis to be wrong given the data. It is true that the smaller the b-value, the stronger the evidence against the evidence against the null hypothesis, but it is not the significance of the test. For the sake of example see Amrhein et al (2017). The earth is flat ($p > 0.05$): significance thresholds and the crisis of unreplicable research, Peer J., 5, e3544, doi:10.7717/peerj.3544.

* Fourth, the Kolmogorov Smirnov test is used and described in a misleading way. The Kolmogorov-

Smirnov test has two versions: i) comparing one distribution with a theoretical distribution (one-sample KS); ii) comparing two sample distributions (KS2). In both cases the test compares two cumulative distributions (one sample versus one theoretical cumulative on KS-one sample; two empirical cumulative distributions for KS-two samples). It is not clear how this is done in this paper. Moreover, it seems to me that the data of these cumulatives are not sampled independently because the data are calculated in strongly overlapped time windows. This creates a problem in applying the KS test, which assumes that the cumulatives are built with independent observations (such as the ones deriving from non-overlapping time windows).

In essence, a paper like this would be much more convincing if the patterns found in a dataset are tested using another independent dataset (like the future events). As is, the conclusions seem weakly supported by the testing phase.

Reviewer #2 (Remarks to the Author):

1. General Comments:

(Only my general comments are given here. For more specific suggestions and comments on the text and figures, please see the second section)

Firstly, I found this submission very interesting and enjoyable, and think that it highlights the potential of using different, but relatively quick and easy to calculate, properties of timeseries data to help recognise eruptive precursors in seismic data - albeit for certain types of eruption/volcanic systems (phreatic).

The submission is of a high standard, and is very clearly and logically written and generally easy to follow. I have no issues with the structuring or language.

I think you have some good and extensive dataset(s) that are a solid basis for conducting this type of study. You make a brave attempt to generalise results from a specific case (Whakaari) to other volcanoes but I actually found this the weakest area, and I have some concerns regarding the appropriateness of using cross-correlation in the way you do to compare precursors at different systems (see detailed comments). I find the section that focuses on Whakaari much stronger and you present a compelling and plausible model to explain the activity and its relationship to the seismic parameters you derive for this case. But I think care should be taken not to overstate the results, and also to maybe to stress that your results only really apply to phreatic eruptions at volcanic systems with persistently restless hydrothermal systems. Maybe you also could include some (short) discussion of how/whether your approach could be extended for (larger) magmatic eruptions?

Another final comment I have is that generally the Methods section is a bit unclear in places, and lacking some details. This is possibly just a consequence of the formatting requirements for this particular type of short article, but I did not find it easy to understand and follow all of the details. Although having said that, you do a good job of giving an overview that explains the big picture more broadly.

###

2. Specific comments:

ABSTRACT

L15 - what are medium/low frequencies? How are these determined?

INTRODUCTION

L28 - is this universally true?

L38 not just tremor - gas driven events are often associated with VLP (very long period) seismic signals - did you observe/look for any in your dataset(s)?

L51 "uses" -> "use"

L53 These frequency values seem arbitrary and case specific? They are mostly based on observations at Whakaari? How much tuning is required? Is this exclusively for andesitic/silicic systems? What about tremor in more basaltic cases?

L55 Maybe needs a tiny more explanation of DSAR (I know it's in the methods section, but another sentence or so here would help)

L63 - here you describe "2-day smoothed and normalized DSAR", but this smoothing is not explained in the Methods section? Unless this refers to the fact that your timeseries are divided into 48 hour windows? If this is the case, maybe make it clearer that this is what you mean.

RESULTS

In terms of identification of the archetypal patterns - e.g. Fig 1 - what happens BEFORE your 4 week periods? How long are the repose periods? Are they similar for each eruption/system?

L69 "potentially correlated" !! You need to clarify/quantify this.

L72/3 - this is speculation... stated as fact.

Why would you necessarily expect correlation? As you say - timescales and magnitudes are different, and processes, material properties, seismic velocities, instrument type, path effects, etc. etc. could be very different. I'm actually surprised the results/correlation are as good as they are - in the case of the Whakaari 2019 and Veniaminof 2013 comparison (see similar point regarding frequencies of the tremor bands). I would maybe expect general patterns to be similar but not to the extent that the signals correlate well (as this is so reliant on the *timescales* matching). Maybe there is an alternative way to compare the features/patterns between different systems? i.e. including/allowing for temporal stretching?

L143 process -> processes

L146 "forth" -> "fourth"

METHODS

L177/8 - So this is all based on ONE station at each volcano? (I assume you just used the vertical components [if any were 3-comp/broadband])?

Did you look at any other stations? Can you be sure your data is representative of the pattern across each volcano's seismic network? i.e. did you consider the possibility of strong site effects for example?

L184 It's a bit unclear what you mean here - if I understand correctly the MF and HF RSAM/amplitude timeseries were (numerically?) integrated (the velocity->displacement transformation) then the ratio calculated? Could this be explained/expanded upon on slightly here for clarity?

L185 - you don't give much detail here on your "outlier detection". Is this based on existing

earthquake catalogues? Or are discrete events/transients removed automatically from the continuous data using "outlier detection"? I think this could be explained better, as I'm not sure your results are reproducible without a little bit more detail here...

L202 - this seems somewhat esoteric - how were the 0.4 and 0.6 values decided upon? I assume this follows an example of previous studies? What happens if you just use the entire range? I'm still also slightly unclear what "paired differences" refers to? The difference between pairs of consecutive values? If so, this is a bit like a numerical differentiation I guess - hence "rate"?? I think this could be explained more clearly.

L204 - why just use the real part? Surely the absolute value would be more intuitive - corresponding to a more usual spectral amplitude at this frequency? And why 75 minutes? I assume this is again tuned/based upon on observations at Whakaari? Or is it completely arbitrary? It's not immediately clear to me why you would expect the period of any oscillations/cyclicity to be *exactly* the same for other volcanic systems - as there are surely so many different controlling variables. So I'm not sure how useful it is to compare this between different volcanoes?

L207 - you don't explicitly state what "nDSAR median" means - but I am assuming it is simply the median value of the normalised DSAR in each 48 hour window?

L221 - "(see Figure 2c-d;" - Is this the wrong figure reference?? I assume you mean Fig3a-c??

FIGURES

Figure 1

In the timeseries panel - which y-axis refers to which? I assume the RHS axis is the raw DSAR and the LHS is the nDSAR?

Figure 2

Replace SX? in caption with relevant fig number...

Figure 3

This is initially a very confusing figure, and it took me a while to understand exactly what it was showing. This is an interesting way of presenting the data - a more traditional approach would be similar to identifying families or multiplets of seismic events using CC, i.e. you slide your template across the longer timeseries and using a threshold value for the CC coeff, identify matching events/times. Did you also try this approach? It seems you have some values with very high coefficients, e.g. above 0.9 for nDSAR - did you cut out/look at these time periods? I assume if a time/value is not labelled then it lies in a repose period?

Maybe as an alternative you could plot the cross-correlation function(s) - i.e. the cross correlation coefficients vs time?

Reviewer #3 (Remarks to the Author):

What are the noteworthy results?

The search for precursors to volcanic eruptions is something of a holy grail in volcano monitoring. While some volcanoes have shown classical elevated seismic and other monitoring signals prior to eruptions that could straightforwardly signal the likelihood of an impending eruption, others, that notably include Whakaari and other volcanoes mentioned in the article, have shown more equivocal precursors. It is becoming increasingly clear, particularly with high-rate data sets such as seismic, that raw data can contain 'hidden' features that are potential precursors. The results presented in this manuscript show that finding and working with these

features might prove very useful to volcano monitoring.

Will the work be of significance to the field and related fields? How does it compare to the established literature? If the work is not original, please provide relevant references.

The work is extending and combining that published by two of the authors (Dempsey and Caudron), one on feature extraction from seismic data and its use in eruption forecasting, and the other on DSAR as an estimator of pressurisation in volcano-hydrothermal systems.

The work will have significance to volcano-seismology, particularly at volcanoes where tremor is commonly observed, and where complex variations in volcanic tremor are a regular occurrence.

I am not as familiar with the literature on complex feature extraction from time-series data sets as I would need to be to comment on the the works relationship to established literature, but I have the feeling it must be leading that work to some extent. The authors do not provide references to similar studies already published so I am assuming they don't exist. If they do, then adding them to the manuscript would support the overall approach of the work.

Does the work support the conclusions and claims, or is additional evidence needed?

The manuscript explains the work and provides sufficient support to the conclusions and claims made. The reader will need to study the supplementary material in some detail as a lot of the background cannot be accommodated in the four figures in the main article. Some of the figure captions in the supplementary material were adequate, but others were a little short and don't describe the figures well enough, Figure S6, for example.

Are there any flaws in the data analysis, interpretation and conclusions? - Do these prohibit publication or require revision?

Smoothed data sets have been used extensively in previous papers on DSAR, with windows of several months. I have always felt this completely altered the nature of the DSAR data and placed some limitations on the usefulness. In this case, the window is just 48 hours, but the smoothing still changes the appearance of the data extensively (Figure 2b), from a series of 'spikes' to a smooth trend which is later compared to other smoother data. I wonder what is the physical interpretation of such extensive smoothing to a data set such as this? This is important in terms of explaining what the feature data sets mean if they are ever used for an operational monitoring system used to inform public actions.

I don't think my concerns about smoothing prohibit publication, but some consideration of this in explaining how the data are constructed would be beneficial.

Is the methodology sound? Does the work meet the expected standards in your field?

The methodology is good. I particularly like how the authors looked at the statistical significance of there results. An earlier paper proposed features that gave a high 'false trigger rate', but this work has directly addressed that by ensuring the pre-eruption signals do not commonly occur in background and general unrest periods.

Is there enough detail provided in the methods for the work to be reproduced?

The code is provided on GitHub so should be enough detail.

Remarks to the Author

These are preceded by line numbers.

25. What is a 'sudden' eruption? Something with a fast onset? But that covers most eruptions. I'm not sure that 'sudden' adds anything to the sentence.

25. Reads better 'Because of this history, and despite history and real-time'

42. Comments about tremor being difficult to distinguish, etc does not apply to Whakaari prior to the 2019 eruption (as shown in the work), and generally at Ruapehu which is also covered in this work.

52. The use of 2-5 Hz filtered for RSAM references two papers. Is the use of 2-5 Hz in this work a result of those references, or for other reasons? I know that GeoNet uses a 2-5 Hz filter for its RSAM data at Whakaari.

66. The word 'fatal' was because people were present at Whakaari when it erupted, not because there was a seal. Request 'fatal' is removed.

71. What does the 'n' in 'nDSAR' mean? I was unable to find an explanation/definition in the text. 71 or thereabouts. I know that sealing at volcanoes can be caused by different processes and that these can have different timescales. The kind of timescale referred to here and elsewhere is on the short end of some I've heard about. Somewhere in the manuscript it would be valuable to talk about sealing time scales. Does that speed of sealing require certain temperature, chemistry, pressure, etc conditions and do those conditions exist at Whakaari, and other volcanoes discussed here?

75. Is the nDSAR median peak is for smoothed nDSAR, if so this needs to be stated. I couldn't find a definition of nDSAR median at all. Lines 202 and 204 explain the other two quantities, but not the median. The meaning of median is obvious, but exactly what it means in this case needs to be defined.

97. Reference to Figure S5. Move the legends in this figure as they interfere with important data in the plots.

108. This paragraph on 75-minute nHF harmonic is great to see. Despite this paragraph, this feature is very obviously shown in Figure 4a (referred to in line 151). Why is it there when it is not differentiable from episodes during repose? Please remove it from Figure 4a.

126. The oscillations were also evident in RSAM.

202. Paired differences of what? This reflects on earlier comments about the need to have a clear description of what these features mean if this work is ever to become operational. Features are difficult to picture as they are complex modifications of specific aspects of data streams.

210. Grammar problem.

223, Correct table is S3.

231. GeoNet has two FDSN data services, the only shown, which is only for the last 8 days of data, and <https://service.geonet.org.nz>, which is for all data 8 days old and older.

Table S2. Ruapehu 2009 July 13 has a * indication, but this is not explained. What I think this means is that this is only an 'eruption' if the meaning of eruption is stretched. GeoNet's Volcanic Alert Bulletin following this 'event' describes a magnitude 2 volcanic earthquake followed by a rise in the level of the volcanic crater lake by 15 cm. GeoNet did not describe this as an eruption. While there was probably a release of fluid from the upper part of the hydrothermal system into the lake, I'm not aware of sealing being discussed at this time, or associated with a similar event in 2021 July 03.

Figure S10. This is not referenced. It covers the period after the 2009 eruption and it's not clear why it is here.

We are grateful for the thoughtful comments that the reviewers have taken time to provide. We have addressed these in the document below according to the following scheme:

Reviewer comment is italicised

Our direct response is normal font and in blue. The text changes are also tracked in the revised manuscript document.

Line xx: modified text. Lines correspond to the final manuscript (not to the track changes one).

Reviewer #1 (Remarks to the Author):

Review of the paper

Seismic Precursors to the Whakaari 2019 Phreatic Eruption detected in Five Other Volcanoes

by Alberto Ardid, David Dempsey, Corentin Caudron, and Shane Cronin

The paper addresses an important scientific and societal issue, i.e., the forecast of gas-driven small eruptions that can cause casualties due to the difficulty in finding good precursors (sometimes these events are named "blue-sky" eruptions).

Although I am overall positive about the goal and the work made by the authors (the paper is well written and clear), I think that the conclusions are weakly supported by the analyses. I hope that this review may help the authors in their work.

My main concern is about the statistical testing phase which seems to be poorly described, rather weak, and maybe wrong in some aspects.

Below I summarize below my main concerns.

- CALCULATING THE PEARSON CORRELATION COEFFICIENT. The use of the Pearson correlation may induce spurious correlations when correlating time series that do not have a white spectrum (e.g., if they have a trend or if they are autocorrelated). In practice, we can find a high correlation between any time series with increasing trend no matter if the trends are due to completely different causes. In this case, the correlation coefficient does not represent a similarity between two time series. Moreover, the Pearson correlation is a parametric measure, and it is not clear its meaning when the data are not distributed according to a normal distribution. These are not necessarily problems, but it could be good to check the autocorrelation of each dataset. If this is not a white noise, it is important to explain why the trend in the dataset has to be taken into consideration in calculating the correlation. Moreover, the use of a nonparametric correlation coefficient like Kendall (or Spearman).

This is a good point, although I think a key aspect as it pertains to our work, is that we focus exclusively on relative correlation. For instance, when checking for recurrency, we rank features on the basis of bulk correlation, and pay particular attention to those with the highest values. When testing differentiability, we are testing the correlation values between eruptions relative to correlation values from non-eruptive data.

We have also performed the analysis using Spearman and Kendall correlation coefficient. The results are similar, because we are looking for the relative differences between

correlations values. We have added a note to this effect in the methods section, and an additional figure in supplementary material (S12).

Furthermore, when initially screening the correlation coefficient matrices for all features, we selected the most prominent features on the basis of both correlation coefficient and visual inspection of the signals. Some of the correlations were spurious, and this was shown up during the subsequent testing phase. Thus, the correlation matrices are only a first guide for selecting the features to be later tested as precursors.

Line 270: 'We also calculated Kendall and Spearman correlations, obtaining similar results (Figure S12; notice the correlation matrices are only a first guide for selecting the features to be later tested as precursors).

- THE SIGNIFICANCE LEVEL OF THE TEST. This is a very important part of the paper, but I think that it is not properly treated by the authors.

* First, if the authors do many tests, the significance level of the test has to be modified accordingly taking into account of the multiple tests. In other words, the significance level of the test makes sense if we perform the test once. If we repeat the test with some variations, or using different eruptions, the significance level of the test is not anymore, the one that has been declared. I do not deepen further this issue that is very well described in statistical literature.

The reviewer makes a good point here (and further down), that we could improve development and application K-S test statistics. However, extending the analysis to compare against arbitrary p-value thresholds invites an unnecessary degree of statistical philosophy into the paper. Really, what we are seeking to demonstrate is that there are measurable distributional differences when comparing eruptions amongst each other, as opposed to comparing them to non-eruptive data. Focusing thus on relative comparison, we feel it is important to compute test statistics for the two different cases and re-phrase the discussion on the relative differences. Justifying meaning to their magnitudes via selection of p-value thresholds is not essential to the main points of the paper.

Following the reviewer's suggestion, we developed three statistical multiple-tests to quantify the significance of the results more robustly. The tests are designed to cross-validate between different eruption subsets to check how broadly the precursors are shared. They also introduce random offsets to check how robust they are (and to guard against false discovery).

*Second, the authors use for the tests the patterns that have the highest correlations with other eruptions. This choice can be done if we aim to find the "common patterns", but it makes very complicated to test the statistical significance, because we are arbitrarily making a cherry picking of the patterns that have to be tested. This problem is pervasive in many fields, and it is the main motivation because models/patterns are estimated in a subset of data and tested (once) in another independent subset. This is not what the authors did (the authors compare the pattern of one single eruption with the others, leading unavoidably to the multitesting problem discussed before); and the data for the testing have also been used to find the precursory patterns.

We understand the reviewer's concern. However, we believe that we are approaching those issues via the following arguments:

- We identified patterns that had the highest correlation amongst all eruptions. We then tested these patterns on the non-eruptive data to check that they were indeed discriminating

of eruptions. In this sense, we have tested in an independent subset of the data. We also show a case where a feature that correlates well across eruptions, but is relatively poor at discriminating when it comes to non-eruptive data (e.g., HF 75-min harmonic (Figure 3c) and the DSAR median for discriminating magmatic eruptions (Figure 3f)). Hence, we have built in tests to check for spurious associations, or false discovery and demonstrated at least one case where this was identified.

With respect to the reviewer, we argue that our approach does not fit a definition of cherry-picking. Cherry-picking is the deliberate exclusion of data that does not support a hypothesis. The hypothesis we have introduced, as a result of our analysis, is that there is a specific pattern in DSAR median and rate variance that occurs prior to eruptions. We have used all data in supporting this hypothesis, not a subset of it. We hope this is clearer now in our expanded discussion on "differentiability" and our testing of the proposed archetypes (and perturbations on them) on over 40 years of eruptive data.

Both cherry-picking and brute force pattern searching (the latter of which we have done here) involve searching and testing many data combinations. A key difference, we feel, is that we have built in safeguards to reduce false detection.

*Third, the p-value is wrongly introduced and interpreted. There is a wide literature on the "p-value hacking", which is based on the wrong interpretation of the p-value. In essence, the p-value is not the "significance of the test", because the p-value is the probability to observe the data given the null hypothesis, whereas the second is the probability of the null hypothesis to be wrong given the data. It is true that the smaller the b-value, the stronger the evidence against the evidence against the null hypothesis, but it is not the significance of the test. For the sake of example see Amrhein et al (2017). The earth is flat ($p > 0.05$): significance thresholds and the crisis of unreplicable research, Peer J., 5, e3544, doi:10.7717/peerj.3544.

We appreciate the reviewer's concern fully. This was lax terminology on our part. We modified the syntax on the main text to discuss instead test statistics.

*Fourth, the Kolmogorov Smirnov test is used and described in a misleading way. The Kolmogorov-Smirnov test has two versions: i) comparing one distribution with a theoretical distribution (one-sample KS); ii) comparing two sample distributions (KS2). In both cases the test compares two cumulative distributions (one sample versus one theoretical cumulative on KS-one sample; two empirical cumulative distributions for KS-two samples). It is not clear how this is done in this paper.

We are using only two-sample Kolmogorov-Smirnov test in the multi testing (which is now explained in the text) to calculate test statistics.

Moreover, it seems to me that the data of these cumulatives are not sampled independently because the data are calculated in strongly overlapped time windows. This creates a problem in applying the KS test, which assumes that the cumulatives are built with independent observations (such as the ones deriving from non-overlapping time windows).

This is a good point. To check if this is an issue, we performed the same statistical tests to the ones presented (Figure 3d-f; Results ->Transferability of eruption precursors), but with

non-overlapping windows when comparing the archetypes with the extended records (Figure S13). A drawback of this approach is that the distribution of correlations values compared with the observed during eruptions has much fewer values. Consequently, a certain sensitivity is noticed when moving the dates that mark the start and end of the non-overlapping windows. On the other hand, the approach presented in the main text with overlapping windows proved more stable. Despite these differences, we obtained quite similar results, indicating that any non-independence is not affecting the conclusions.

Line 318: The two-sample K-S test presumes samples contain independent observations. In practice, correlation coefficients computed on non-eruptive windows can have significant overlap. We checked if this was affecting our results by recalculating the tests using fewer, non-overlapping windows for non-eruptive data (Fig. S13). The results are not materially different to Fig. 3.

In essence, a paper like this would be much more convincing if the patterns found in a dataset are tested using another independent dataset (like the future events). As is, the conclusions seem weakly supported by the testing phase.

The suggestion to test on a future dataset is obviously a gold standard but is impractical as a requirement because: (a) the next eruption may not occur for years or decades - should the research be indefinitely delayed? (b) publication in the open scientific literature enables the hypothesis advanced here to be publicly "on record" and readily testable when the next eruption does occur.

However, we hope the revised testing regime has addressed your concern about the strength of support for these precursors.

-

We are grateful to the reviewer for the constructive comments, and we have made a significant effort to clarify and expand most of the issues raised.

Reviewer #2 (Remarks to the Author):

1. General Comments:

(Only my general comments are given here. For more specific suggestions and comments on the text and figures, please see the second section)

Firstly, I found this submission very interesting and enjoyable, and think that it highlights the potential of using different, but relatively quick and easy to calculate, properties of timeseries data to help recognise eruptive precursors in seismic data - albeit for certain types of eruption/volcanic systems (phreatic).

The submission is of a high standard, and is very clearly and logically written and generally easy to follow. I have no issues with the structuring or language.

I think you have some good and extensive dataset(s) that are a solid basis for conducting this type of study. You make a brave attempt to generalise results from a specific case (Whakaari) to other volcanoes but I actually found this the weakest area, and I have some concerns regarding the appropriateness of using cross-correlation in the way you do to compare precursors at different systems (see detailed comments). I find the section that focuses on Whakaari much stronger and you present a compelling and plausible model to

explain the activity and its relationship to the seismic parameters you derive for this case. But I think care should be taken not to overstate the results, and also to maybe to stress that your results only really apply to phreatic eruptions at volcanic systems with persistently restless hydrothermal systems. Maybe you also could include some (short) discussion of how/whether your approach could be extended for (larger) magmatic eruptions?

We recognize the reviewer's concern. Following reviewer one advice (see first answer to reviewer one) and reviewer two, we performed several statistical multi tests to explore and quantify the extent to which the proposed precursors located in Whakaari are transferable the other volcanoes.

Results suggest good transferability of DSAR median and rate variance within eruptions at Whakaari, and reduced transferability to Ruapehu and Tongariro (all three usually present phreatic eruptions). Results also indicates that neither precursor is confidently transferable to the magmatic eruptions at the other volcanoes.

Testing these three subsets (see new section 'Transferability of eruption precursors') has helped us understand the limits of precursor transferability between systems. While this is limited, it still has value to be reported. As suggested, we have walked back the claim that these precursors are universally applicable, but instead focus on the volcano subsets they do apply to, and why this should be the case.

Another final comment I have is that generally the Methods section is a bit unclear in places, and lacking some details. This is possibly just a consequence of the formatting requirements for this particular type of short article, but I did not find it easy to understand and follow all of the details. Although having said that, you do a good job of giving an overview that explains the big picture more broadly.

Thanks. We have extended the methods section in places to try and make the ideas easier to follow.

###

2. Specific comments:

ABSTRACT

L15 - what are medium/low frequencies? How are these determined?

Medium (MF; 4-8 Hz) and high-frequency bands (HF; 8-16 Hz). Methods section has been expanded. More details on this band selection can be followed in Caudron et al., 2019 and Dempsey et al., 2020.

Line 14: 'The precursor manifests in the displacement seismic amplitude ratio between medium (4.5-8 Hz) and high (8-16 Hz) frequency tremor bands...'

Line 228: 'The 4-5 Hz threshold between RSAM and MF reflects an assumption that tremor mostly radiates energy below 4.5 Hz. To exclude this effect and explore attenuation related to permeability change (such as sealing), this frequency value is used as a threshold.'

INTRODUCTION

L28 - is this universally true?

Statement softened.

Line 29: 'Phreatic eruptions are usually driven by expansion and steam-flashing of suddenly released hot pressurized fluids (e.g., Montanaro et al., 2016).'

L38 not just tremor - gas driven events are often associated with VLP (very long period) seismic signals - did you observe/look for any in your dataset(s)?

Our vocabulary wasn't precise there. We change it to 'seismic signals' to make it more general. We haven't look for VLP in our dataset. We will eventually, but we don't think is relevant for the study presented as we are removing outliers from the seismic signal.

Line 46: 'Continuous volcanic seismic signals are interrogated to understand fluid movement (e.g., magma degassing), conduit fracture/bubble processes that precede eruptions, and pressurization of a hydrothermal system (Gottsmann et al., 2007; Ohminato et al., 2006; Caudron et al., 2019).'

L51 "uses" -> "use"

Modified

L53 These frequency values seem arbitrary and case specific? They are mostly based on observations at Whakaari? How much tuning is required? Is this exclusively for andesitic/silicic systems? What about tremor in more basaltic cases?

The cutoff is based on the assumption that tremor mostly radiates energy below 4.5 Hz. To exclude this effect and explore attenuations due to permeability changes (as sealing), we select this frequency as a threshold. More details on how they were selected are given in Caudron et al. 2019 (<https://doi.org/10.1130/G46107.1>). A worthwhile next step in this research is the selection of features in frequency space that don't require the use of fixed frequency bands.

Line 228: 'The 4-5 Hz threshold between RSAM and MF reflects an assumption that tremor mostly radiates energy below 4.5 Hz. To exclude this effect and explore attenuation related to permeability change (such as sealing), this frequency value is used as a threshold.'

L55 Maybe needs a tiny more explanation of DSAR (I know it's in the methods section, but another sentence or so here would help)

Extended. Also, the methods have been extended and a quick recipe on hoe to calculate it is given in the supplemental material.

Line 65: 'The displacement seismic amplitude ratio (DSAR; MF/HF) has been linked to month- and year-long variability prior to eruptions (Caudron et al., 2021) that is interpreted as progressive sealing and tendency to pressurization. Rather than using earthquakes, DSAR has been used to investigate changes in the persistent shallow seismic vibrations by adopting this simple band-ratio approach (Caudron et al., 2019).'

L63 - here you describe "2-day smoothed and normalized DSAR", but this smoothing is not explained in the Methods section? Unless this refers to the fact that your timeseries are divided into 48 hour windows? If this is the case, maybe make it clearer that this is what you mean.

Apologies, this was a shorthand way to describe how the feature was calculated. But in the end, it reduced clarity more than it save word space. We have corrected to a '2-day normalized DSAR tremor'

Line 76: 'We identified characteristic peaks in the 2-day median of normalized DSAR tremor (nDSAR median; see Methods; n denotes normalization) that...'

RESULTS

In terms of identification of the archetypal patterns - e.g. Fig 1 - what happens BEFORE your 4 week periods? How long are the repose periods? Are they similar for each eruption/system?

The selection of a 4 week window was to focus the search on short-term precursors that could assist short-term forecast resolution in the days and weeks before eruptions. Long term evolution and unrest is probably important as well, although not straightforward to quantify and incorporate (although we'd like to get there!)

Monitoring systems in New Zealand have been successful in detecting precursory activity on the scale of months before eruptions (for example in the recent Whakaari 2019 where volcanic tremor was detected 2 months before the event, that increased 1 month before the eruption, but no other relevant activity was reported later in the 4 weeks before the eruption). In any case, we tested the correlations for 3 and 5 weeks before eruptions, and no significant differences were noticed, so we decided on four weeks (also supported by findings for Whakaari 2019; Figure 4). The correlation is used to guide the posterior selection of potential precursors, so the four-week choice could be flexible without modifying the resulting selection.

Next steps in our research consider extending the analysis on the scale of months to years.

L69 "potentially correlated" !! You need to clarify/quantify this.

Corrected ('potentially' removed)

Line 83: 'In the four weeks leading up to an eruption, we observe correlated patterns across at least three classes of derived tremor time series.'

L72/3 - this is speculation... stated as fact.

Corrected.

Line 87: 'This feature quantifies variance in the "spikiness" of nDSAR, however, its pattern is not obvious from visual inspection of the raw data.'

Why would you necessarily expect correlation? As you say - timescales and magnitudes are different, and processes, material properties, seismic velocities, instrument type, path effects, etc. etc. could be very different. I'm actually surprised the results/correlation are as good as they are - in the case of the Whakaari 2019 and Veniaminof 2013 comparison (see similar point regarding frequencies of the tremor bands). I would maybe expect general patterns to be similar but not to the extent that the signals correlate well (as this is so reliant on the *timescales* matching). Maybe there is an alternative way to compare the features/patterns between different systems? i.e. including/allowing for temporal stretching?

An important point that we haven't stressed well is that we are demonstrating *relatively* high correlation, as opposed to absolute correlation. There will be some offsets and stretches, and these will erode the level of correlation – but not to the extent of uncorrelated signals (which we see when comparing to random stretches of non-eruptive data).

We agree that general correlations between eruptions are likely the best possible. Indeed, the patterns we report are deliberately not especially detailed and specific, in order to generalize the “archetype” concept to some bulk quality shared amongst a set of eruptions.

From our analysis, we now consider that a straight correlation coefficient is not the best measure of similarity for exploring common feature patterns. It may be too sensitive to minor offsets of the signals or stretches in time (stated now in the main text; Methods). Although we think it is adequate for this study, in the future there are some more sophisticated alternatives such as maximum convolution or dynamic time-warping.

Line 275: 'The correlation method is sensitive to temporal offsets of signals or stretching on the time axis. The former is partially compensated by introducing a “jitter” at multi-testing. However, the method is only used for initial screening of potential precursors and spurious correlations are identified at the next stage of statistical testing. Sophisticated alternatives such as maximum convolution or dynamic time-warping may reveal other patterns missed by our approach.'

L143 process -> processes

Corrected

L146 "forth" -> "fourth"

Corrected

METHODS

L177/8 - So this is all based on ONE station at each volcano? (I assume you just used the vertical components [if any were 3-comp/broadband])?

Yes. Feature time series are calculated from one component (vertical) in one station per volcano (Table S1).

Did you look at any other stations? Can you be sure your data is representative of the pattern across each volcano's seismic network? i.e. did you consider the possibility of strong site effects for example?

Criteria for selecting stations was based on (1) was it active over the eruptions of interest and (2) was it close to the volcano source. We manually checked signals from several stations over the networks around the volcanoes, and selected the most suitable ones given their signal to noise ratios.

For most of the NZ volcanoes, justification is on the basis of a site having adequate temporal coverage across multiple eruptions. Site effects may well be present (that will undermine our ability to locate transferable precursors), and this is one reason it is important to normalize the data before attempting any sort of comparison.

In the end, we hope this problem might be overcome by increasing the scale of the study. By including more and more volcanoes/eruptions, we decrease the weight of individual stations,

and especially those with difficult site characteristics.

L184 It's a bit unclear what you mean here - if I understand correctly the MF and HF RSAM/amplitude timeseries were (numerically?) integrated (the velocity->displacement transformation) then the ratio calculated? Could this be explained/expanded upon on slightly here for clarity?

DSAR is computed by integrating the MF and HF signals. The methods section have been extended for making these steps clearer. RSAM, MF and HF are computed by first filtering on day signals in the corresponding frequency bands, then computing the average absolute signals over 10-min windows, and for the DSAR taking the ratio between the integrated quantities MF and HF (more details can be checked in Caudron et al, 2019). Also, a quick recipe to calculate the data streams are given in the supplemental material.

L185 - you don't give much detail here on your "outlier detection". Is this based on existing earthquake catalogues? Or are discrete events/transients removed automatically from the continuous data using "outlier detection"? I think this could be explained better, as I'm not sure your results are reproducible without a little bit more detail here...

Method section extended. Wave trains produced by regional earthquakes temporarily elevate the velocity record at seismic stations above the background tremor. When averaged over a 10-minute interval, this produces outlier RSAM values that confuse algorithms with no contextual understanding (human operators are quick to diagnose and dismiss this noise). So we needed a method that could identify and remove earthquake signals from the RSAM/MF/HF computation. We chose not to use official catalogues, as this would create a reliance on an external service with a potential lag time.

To identify wave trains, we applied outlier detection to each 10-minute seismic velocity trace and excised any anomalous signals (see methods). Other methods to automatically identify earthquake signals (i.e., Mousavi et al., 2020) could also be used.

Line 231: 'To isolate continuous tremor signal, outlier detection was used to remove regional and volcano-tectonic (VT) earthquakes before each of the above data streams was calculated. Wave trains from earthquakes perturb the average velocity record at stations above the background. After averaging, there is an outlier in the RSAM that adds noise and confuses algorithms without context. To identify and remove the earthquake effects, we applied an outlier detection algorithm to raw velocity traces, excising a two-minute envelope whenever velocity exceeded three standard deviations from the data 10-min window mean. This removed most short duration events but still resulted in contamination by larger subduction earthquakes. To remove these, we applied a two-window (20 minutes), moving-minimum. Other methods to automatically identify earthquake signals (Mousavi et al., 2020) could also be used.'

L202 - this seems somewhat esoteric - how were the 0.4 and 0.6 values decided upon? I assume this follows an example of previous studies? What happens if you just use the entire range? I'm still also slightly unclear what "paired differences" refers to? The difference between pairs of consecutive values? If so, this is a bit like a numerical differentiation I guess - hence "rate"?? I think this could be explained more clearly.

Yes this was not well communicated well in the text. We improve it to make it clearer.

tsfresh library calculates around 700 features. As subsets of features, there are 'families' of features that capture similar characteristics. For example, the 'median' has strong similarities

with other percentile feature time series calculated from tsfresh. When we developed the correlation analysis, we obtained very similar results (or patterns) for DSAR feature time series of percentiles 0.3, 0.4, 0.5 (median), 0.6 and 0.7. For simplicity and better communications purpose, we decided to show the median (0.5 percentile), because for most readers will be easier to understand. The case is similar to the other two features presented. For the DSAR rate variance, that we named after 'change_quantiles__f_agg_"var"__isabs_False__qh_0.6__ql_0.4' feature, case is similar as it presents similar patterns to equivalent change quantiles ranges (i.e., ~.3-.7) but we decided to keep and show one for simplicity. Same for the 75-minute nHF harmonic, named after 'fft_coefficient__coeff_38__attr_"real"' feature, that present similar patterns for several fft coefficients time series (~30-60) in both real and imaginary components. Again, for simplicity, we decided to show just one representative of the subset.

Line 261: 'The tsfresh library calculates many features. However, there are 'families' of features that capture similar characteristics and are highly correlated with one another. For example, the 'median' feature (i.e., Figure 2) has strong similarities with other time series quantiles. When undertaking eruption correlation analysis, we obtained quite similar results for DSAR quantiles 0.3, 0.4, 0.5 (median), 0.6 and 0.7. We decided to use the median (0.5 quantile) as it is more understandable. A similar situation occurs for the DSAR rate variance (that measure variance of adjacent data differences falling in the interquartile range 0.4-0.6), that has quite similar patterns across other quantile ranges (i.e., ~0.3-0.7).'

L204 - why just use the real part? Surely the absolute value would be more intuitive - corresponding to a more usual spectral amplitude at this frequency?

And why 75 minutes? I assume this is again tuned/based upon on observations at Whakaari? Or is it completely arbitrary? It's not immediately clear to me why you would expect the period of any oscillations/cyclicity to be *exactly* the same for other volcanic systems - as there are surely so many different controlling variables. So I'm not sure how useful it is to compare this between different volcanoes?

The 75-minute nHF harmonic appears in several eruptions from Whakaari and Ruapehu, Bezymianny, Veniaminof and Pavlof. As mentioned in the previous answer, equivalent strong correlations were observed in similar harmonics time series, real and imaginary, but we decided to show one representative of the collective. However, later we found that these harmonics that highlight high-frequency activity are not precursors, because they occurred not only before eruptions (as shown in Figure 3c). Even if they are not helpful to compare different volcanoes, the example is useful for showing what is not a precursor, but might look like one. Accordingly, it is not used later, especially when interpreting the Whakaari 2019 eruption (Figure 4a).

L207 - you don't explicitly state what "nDSAR median" means - but I am assuming it is simply the median value of the normalised DSAR in each 48 hour window?

The nDSAR median correspond to a time series calculated as the 'median' from the normalized DSAR data subdivided into overlapping 48-hour windows every ten minutes. Also, we added a Quick recipes section in the supplemental material on how to calculate both features of interested.

Line 253: 'The feature nDSAR median corresponds to a time series calculated as the 'median' from the normalized DSAR data in a 48-hour window.' Also, see Quick recipes section in the supplemental material.

L221 - "(see Figure 2c-d;" - Is this the wrong figure reference?? I assume you mean Fig3a-c??

Corrected.

Line 288: 'Histograms are used to summarize the overall distribution and percentiles to identify where individual eruptions fall on the distribution (Figure 3a-c).'

FIGURES

Figure 1

In the timeseries panel - which y-axis refers to which? I assume the RHS axis is the raw DSAR and the LHS is the nDSAR?

Yes. A clarification is added in the caption.

Figure 2

Replace SX? in caption with relevant fig number...

Corrected

Figure 3

This is initially a very confusing figure, and it took me a while to understand exactly what it was showing. This is an interesting way of presenting the data - a more traditional approach would be similar to identifying families or multiplets of seismic events using CC, i.e. you slide your template across the longer timeseries and using a threshold value for the CC coeff, identify matching events/times. Did you also try this approach? It seems you have some values with very high coefficients, e.g. above 0.9 for nDSAR - did you cut out/look at these time periods? I assume if a time/value is not labelled then it lies in a repose period?

We understand the concern. I believe that what you are suggesting (slide the template across the longer timeseries and using a threshold value for the CC coeff, identify matching events/times) is very similar to what we are doing here, as we slide the template across the longer timeseries, and from the statistics calculate 'threshold' that are the percentiles (black lines) shown. Next steps in our research will include locating events over these thresholds to compare and complete catalogues. For the purpose of this paper, we decided to complement the study with more formal statistical tests (as also suggested from reviewer one) (Figure 3d-f).

Maybe as an alternative you could plot the cross-correlation function(s) - i.e. the cross-correlation coefficients vs time?

We tried it, but given the extent of the records, a histogram of the values obtained for each day seemed more reasonable, also because is easier to compare it directly with the values observed before eruptions. We decided to complement this figure, and the overall statistical analysis, with more standard statistical tools. In particular with multi testing using p-values statistics to validate our results and better communication (Figure 3d-f).

-

We thank the reviewer for the extensive and constructive review.

Reviewer #3 (Remarks to the Author):

What are the noteworthy results?

The search for precursors to volcanic eruptions is something of a holy grail in volcano monitoring. While some volcanoes have shown classical elevated seismic and other monitoring signals prior to eruptions that could straightforwardly signal the likelihood of an impending eruption, others, that notably include Whakaari and other volcanoes mentioned in the article, have shown more equivocal precursors. It is becoming increasingly clear, particularly with high-rate data sets such as seismic, that raw data can contain 'hidden' features that are potential precursors. The results presented in this manuscript show that finding and working with these features might prove very useful to volcano monitoring.

Thanks for the positive feedback. I agree that the essence of this study is showing that the raw data contain 'hidden' features with the potential to be monitored as precursors.

Will the work be of significance to the field and related fields? How does it compare to the established literature? If the work is not original, please provide relevant references.

The work is extending and combining that published by two of the authors (Dempsey and Caudron), one on feature extraction from seismic data and its use in eruption forecasting, and the other on DSAR as an estimator of pressurisation in volcano-hydrothermal systems.

The work will have significance to volcano-seismology, particularly at volcanoes where tremor is commonly observed, and where complex variations in volcanic tremor are a regular occurrence.

I am not as familiar with the literature on complex feature extraction from time-series data sets as I would need to be to comment on the the works relationship to established literature, but I have the feeling it must be leading that work to some extent. The authors do not provide references to similar studies already published so I am assuming they don't exist. If they do, then adding them to the manuscript would support the overall approach of the work.

Does the work support the conclusions and claims, or is additional evidence needed?

The manuscript explains the work and provides sufficient support to the conclusions and claims made. The reader will need to study the supplementary material in some detail as a lot of the background cannot be accommodated in the four figures in the main article. Some of the figure captions in the supplementary material were adequate, but others were a little short and don't describe the figures well enough, Figure S6, for example.

Captions in supplementary materials have been reviewed and extended.

Are there any flaws in the data analysis, interpretation and conclusions? - Do these prohibit publication or require revision?

Smoothed data sets have been used extensively in previous papers on DSAR, with windows of several months. I have always felt this completely altered the nature of the DSAR data and placed some limitations on the usefulness. In this case, the window is just 48 hours, but the smoothing still changes the appearance of the data extensively (Figure 2b), from a series of 'spikes' to a smooth trend which is later compared to other smoother data. I wonder what is the physical interpretation of such extensive smoothing to a data set such as this? This is important in terms of explaining what the feature data sets mean if they are ever used for an operational monitoring system used to inform public actions.

Smoothing is likely to be filtering high frequencies including noise and surface activity. Given that usually surface noise is of high amplitude, filtering by smoothing help to highlight lower frequency hidden patterns, like the one in Figure 2b. This can be appreciated in Figure 4: 4d (the raw data for each frequency) the hidden patterns are nearly appreciable, but after the smoothing by calculating the median feature time series, several lower period patterns emerged, as shown in 4b and 4a.

I don't think my concerns about smoothing prohibit publication, but some consideration of this in explaining how the data are constructed would be beneficial.

The feature subsection in methods has been extended given this concern. Also, we added a 'Quick recipes' section in the supplemental material on how to calculate the data streams and the features of interest.

Is the methodology sound? Does the work meet the expected standards in your field?
The methodology is good. I particularly like how the authors looked at the statistical significance of their results. An earlier paper proposed features that gave a high 'false trigger rate', but this work has directly addressed that by ensuring the pre-eruption signals do not commonly occur in background and general unrest periods.

We appreciate the comment. We have extended the statistical significance analysis, which is one of the most innovative aspects of this work, to be more in agreement with traditional statistical tests (see Results, Transferability of eruption precursors).

Is there enough detail provided in the methods for the work to be reproduced?
The code is provided on GitHub so should be enough detail.

Also, the data is freely available.

Remarks to the Author

These are preceded by line numbers.

25. What is a 'sudden' eruption? Something with a fast onset? But that covers most eruptions. I'm not sure that 'sudden' adds anything to the sentence.

'Sudden' removed. Agreed that it is a bit ambiguous.

25. Reads better 'Because of this history, and despite history and real-time'

Rewritten.

Line 26: 'Because of this history, and despite real-time monitoring, identification and early warning of eruption precursors at this and similar volcanoes remains challenging.'

42. Comments about tremor being difficult to distinguish, etc does not apply to Whakaari prior to the 2019 eruption (as shown in the work), and generally at Ruapehu which is also covered in this work.

Agreed. Modified. We were trying to communicate that even though tremors have been detected prior to eruptions, as before the Whakaari 2019 eruption (GeoNet, 2019a), they have been usually indicative of volcanic unrest state but limited for anticipating an eruption in the short term (days to weeks).

Line 51: 'Detection of tremor prior to Whakaari eruptions (GeoNet, 2019a) has helped support diagnoses of volcanic unrest but has not been used to establish eruption imminence. This is partly due to the very shallow origin of these signals, with pressurization and triggering occurring within a few tens to hundreds of metres of the surface (Montanaro et al., 2016; Jolly et al., 2020; Caudron et al., 2021).'

52. The use of 2-5 Hz filtered for RSAM references two papers. Is the use of 2-5 Hz in this work a result of those references, or for other reasons? I know that GeoNet uses a 2-5 Hz filter for its RSAM data at Whakaari.

The two references are for the ones that developed and tested the RSAM (Aki and Koyanagi, 1981; McNutt, 1994). Then we reference some papers that applied it including New Zealand ones (Chardot et al., 2015; Dempsey et al., 2020). We are using that band mainly because it is typically used in New Zealand.

Line 59

66. The word 'fatal' was because people were present at Whakaari when it erupted, not because there was a seal. Request 'fatal' is removed.

Removed.

Line 79: 'Corroborated by other observations, we present a timeline of the 2019 Whakaari eruption, explaining how rapid sealing in the shallow hydrothermal system promoted pressurization and created the conditions for a phreatic explosion.'

71. What does the 'n' in 'nDSAR' mean? I was unable to find an explanation/definition in the text.

'n' indicated normalized. Clarification is now added in the text. Also, see Quick recipes on supplemental material on how to calculate the features of interest.

Line 77: '(nDSAR median; see Methods; n denotes normalization)'

Line 253: 'The feature nDSAR median corresponds to a time series calculated as the 'median' from the normalized DSAR data in a 48-hour window.'

71 or thereabouts. I know that sealing at volcanoes can be caused by different processes and that these can have different timescales. The kind of timescale referred to here and elsewhere is on the short end of some I've heard about. Somewhere in the manuscript it would be valuable to talk about sealing time scales. Does that speed of sealing require certain temperature, chemistry, pressure, etc conditions and do those conditions exist at Whakaari, and other volcanoes discussed here?

Agree. A short discussion on seal formation timescales is necessary.

Line 37: 'Hydrothermal seals and their failure can play an important role in phreatic eruptions because diminished vent permeability drives pressurization whereas a rapid increase can drive sudden decompression (Christenson et al. 2010; Mayer et al. 2015; Rodgers et al. 2015). The timescales of seal formation vary depending on fluid state, chemistry, lithology, temperature and pressure (Stix and de Moor 2018). In hydrothermal systems below crater lakes, sealing varies from weeks to months (Christenson et al. 2017) for mineral precipitation

(sulphur, sulphates, silica). In other cases they may form in seconds to hours (Kennedy et al., 2020) due to mineralized zones (alunite, anhydrite, smectite) that open and close in response to pressure variations.'

75. Is the nDSAR median peak is for smoothed nDSAR, if so this needs to be stated. I couldn't find a definition of nDSAR median at all. Lines 202 and 204 explain the other two quantities, but not the median. The meaning of median is obvious, but exactly what it means in this case needs to be defined.

We added a reference to Feature extraction in Methods where we now extend of the nDSAR median.

Line 253: 'The feature nDSAR median corresponds to a time series calculated as the 'median' from the normalized DSAR data in a 48-hour window.' Also, we also add a subsection at the end of Methods with a quick recipe to calculate it.

108. This paragraph on 75-minute nHF harmonic is great to see. Despite this paragraph, this feature is very obviously shown in Figure 4a (referred to in line 151). Why is it there when it is not differentiable from episodes during repose? Please remove it from Figure 4a.

We used it for interpretation, but following the advice we removed it.

126. The oscillations were also evident in RSAM.

Corrected.

Line 166: 'Short, day-long cyclic oscillations are evident in RSAM, MF and HF bands (Figure 4c).'

202. Paired differences of what? This reflects on earlier comments about the need to have a clear description of what these features mean if this work is ever to become operational. Features are difficult to picture as they are complex modifications of specific aspects of data streams.

Paired differences of nDSAR values falling in the interquartile range 0.4-0.6. This is a tricky feature to interpreted, but it basically corresponds to the variance in the rate between percentile content between .4 and .6. We named it 'nDSAR rate variance' because it is indicative of the rate of change variance of a smoothed version of the DSAR ratio. We also add a subsection at the end of Methods with a quick recipe to calculate it.

223, Correct table is S3.

Corrected.

231. GeoNet has two FDSN data services, the only shown, which is only for the last 8 days of data, and <https://service.geonet.org.nz>, which is for all data 8 days old and older.

Added.

Table S2. Ruapehu 2009 July 13 has a * indication, but this is not explained. What I think this means is that this is only an 'eruption' if the meaning of eruption is stretched. GeoNet's

Volcanic Alert Bulletin following this 'event' describes a magnitude 2 volcanic earthquake followed by a rise in the level of the volcanic crater lake by 15 cm. GeoNet did not describe this as an eruption. While there was probably a release of fluid from the upper part of the hydrothermal system into the lake, I'm not aware of sealing being discussed at this time, or associated with a similar event in 2021 July 03.

As you point, Ruapehu 2009 July 13 was reported as phreatic activity. By inspection of the data available (seismic, lake temperature, lake level, and rainfall data; see new Figure S10), we consider that it might be a missed small eruption given the sharp increase in the lake level (without rain) and a significant peak in the RSAM (and MF and HF data). Given the presence of the lake that could obscure small events, missing small eruptions in Ruapehu is reasonable. By closer inspection, we found that a series of high amplitudes nDSAR median cycles occurred before the event (Figure S2). Based on these attributes, we decided to incorporate it into the analysis. However, we acknowledge that this is not a recognised eruption by GeoNet, and we are stating it accordingly.

Figure S10. This is not referenced. It covers the period after the 2009 eruption and it's not clear why it is here.

Removed.

-

We thank the reviewer for the extensive and constructive review.

REVIEWERS' COMMENTS

Reviewer #1 (Remarks to the Author):

Review of the paper

Seismic Precursors to the Whakaari 2019 Phreatic Eruption detected in Five Other Volcanoes
by Alberto Ardid, David Dempsey, Corentin Caudron, and Shane Cronin

This is a deeply revised version of a paper that I have already reviewed. The paper addresses an important scientific and societal issue, i.e., the forecast of gas-driven small eruptions that can cause casualties due to the difficulty in finding good precursors (sometimes these events are named "blue-sky" eruptions).

I think that the authors did a very good job in addressing my main concerns.

However, I have a few more suggestions that can make the paper clearer and more correct (in my opinion of course)

They are easily addressable, but very important in my mind.

- The last statement of the abstract reports "This method for identifying and proving generalizable eruption precursors can help improve short term forecasting systems."

That's correct, but in my opinion this statement is rather fuzzy and reminds me tons of papers that end their abstract saying "... and this may be helpful for seismic/volcanic hazard", without specifying how.

I think the authors may be a little bit more specific saying that retrospective analyses show that the method show some interesting predictive performances, or something like that.

- Related to the previous point, I agree with the authors that they cannot wait for a decade or more to test their model (as they wrote in their rebuttal letter), but to me it is very important to acknowledge the fact that the results obtained in this paper are encouraging in retrospective analyses but the real forecasting skill of their findings can be measured only in prospective experiments. In my view this does not diminish the importance of the paper, but it implicitly acknowledges the fact that (not necessarily this case) retrospective analyses have been often widely optimistic when tested in real prospective experiments (this fact is pervasive in earth sciences). Again, no new analyses to add, but to me it could be very helpful to highlight the different roles of retrospective and prospective tests and analyses along the paper (and maybe in the abstract). I repeat myself, this does not lower the importance of the paper but, in my eyes, it puts the results in a more scientific context.

I apologize if this comment may sound trivial or "philosophical", but I think that it is much more than that. Acknowledging the difference between retrospective and prospective analyses has a pedagogical value to explain a key issue that is often overlooked and can explain the problem of lack of reproducibility of the results of many papers.

To be fair, this is almost never acknowledged in many papers published in top journals. Hence, of course this is just my personal view.

- The authors use the term "test statistic" in an inappropriate way. In my understanding this is the p-value of the test (for KS2, the test statistics is the difference between the cumulatives, whereas here it appears a probability). I do not want to appear pedantic, but using a correct terminology may be important to make the paper appreciated also by statisticians.

So, if I understood correctly, I suggest to replace "test statistics" with "p-value", or explaining well that in this paper what we mean with the term test statistics. To be clear, in my previous review I have just highlighted the fact the the p-value is not the statistical significance. Here this problem has been solved, because the p-value is used as a quantity to be compared considering eruption periods and quiet phases.

Reviewer #2 (Remarks to the Author):

Thank you for your very detailed comments and responses to my suggestions.

I think you have done a really excellent and thorough job with the changes, and you have pretty much addressed all the main concerns/questions that I had, as well as those of the other reviewers I think, since we had some overlap in our comments - especially in how the results are/can be generalised and transferred to other volcanoes. I think the additional section on transferability has definitely strengthened the overall arguments presented.

Also, I think the changes to the methods section in particular really help, and I am now much clearer on some of the details that were missing or unclear previously, such as the DSAR median and rate variance and outlier removal. The "quick recipes" in the supplementary material also helps clarify the exact processing steps required in order to reproduce the results.

I am happy to recommend and endorse this manuscript for publication

Reviewer #3 (Remarks to the Author):

This is a follow up review after revisions to the manuscript requested in earlier review.

I'm happy that the intent of my comments have been reflected in revisions carried out by the authors. I particularly appreciate the addition of material that better explains how key features were calculated, including quick recipe supplemental section.

line 25/26 the authors noted they'd removed 'sudden' in describing the 2019 Whakaari eruption, but it remains in the text. Maybe it was here twice and only one has been removed???

line 42 and related. The discussion on tremor and sealing is a lot better and I think adds substantially to the logic behind how the authors are interpreting the changes in tremor and derived features.

I would be happy to see the manuscript published in it's current form.

We are grateful for the thoughtful comments that the reviewers have taken time to provide. We have addressed these in the document below according to the following scheme:

Reviewer comment is italicised

Our direct response is normal font and in blue. The text changes are also tracked in the revised manuscript document.

Line xx: modified text. Lines correspond to the final manuscript (not to the track changes one).

REVIEWERS' COMMENTS

Reviewer #1 (Remarks to the Author):

Review of the paper

Seismic Precursors to the Whakaari 2019 Phreatic Eruption detected in Five Other Volcanoes

by Alberto Ardid, David Dempsey, Corentin Caudron, and Shane Cronin

This is a deeply revised version of a paper that I have already reviewed. The paper addresses an important scientific and societal issue, i.e., the forecast of gas-driven small eruptions that can cause casualties due to the difficulty in finding good precursors (sometimes these events are named "blue-sky" eruptions).

I think that the authors did a very good job in addressing my main concerns.

We are grateful to the reviewer for the extensive review provided.

However, I have a few more suggestions that can make the paper clearer and more correct (in my opinion of course)

They are easily addressable, but very important in my mind.

- The last statement of the abstract reports "This method for identifying and proving generalizable eruption precursors can help improve short term forecasting systems."

That's correct, but in my opinion this statement is rather fuzzy and reminds me tons of papers that end their abstract saying "... and this may be helpful for seismic/volcanic hazard", without specifying how.

We have rephrase that sentence in the abstract. We agree that that phrase has been lightly used in the past.

Line 19: 'Real-time monitoring for this precursor may improve short-term eruption warning systems at certain volcanoes.'

I think the authors may be a little bit more specific saying that retrospective analyses show that the method show some interesting predictive performances, or something like that.

- Related to the previous point, I agree with the authors that they cannot wait for a decade or more to test their model (as they wrote in their rebuttal letter), but to me it is very important to acknowledge the fact that the results obtained in this paper are encouraging in retrospective

analyses but the real forecasting skill of their findings can be measured only in prospective experiments. In my view this does not diminish the importance of the paper, but it implicitly acknowledges the fact that (not necessarily this case) retrospective analyses have been often widely optimistic when tested in real prospective experiments (this fact is pervasive in earth sciences). Again, no new analyses to add, but to me it could be very helpful to highlight the different roles of retrospective and prospective tests and analyses along the paper (and maybe in the abstract). I repeat myself, this does not lower the importance of the paper but, in my eyes, it puts the results in a more scientific context.

I apologize if this comment may sound trivial or "philosophical", but I think that it is much more than that. Acknowledging the difference between retrospective and prospective analyses has a pedagogical value to explain a key issue that is often overlooked and can explain the problem of lack of reproducibility of the results of many papers.

To be fair, this is almost never acknowledged in many papers published in top journals. Hence, of course this is just my personal view.

We agreed with the reviewer. We have added a paragraph in the conclusion pointing that.

Line 220: 'Our retrospective analysis suggests these precursors have encouraging properties for short term eruption forecasting. This could include real-time monitoring for individual precursors, or using models that optimally combine them. However, true forecasting skill must be demonstrated with prospective experiments, which underscores how operationalization, testing and post-hoc analysis are essential complements to predictive volcanology. This is the best way to produce trusted forecasts that can improve safety at inhabited or regularly visited volcanoes.'

- The authors use the term "test statistic" in an inappropriate way. In my understanding this is the p-value of the test (for KS2, the test statistics is the difference between the cumulatives, whereas here it appears a probability). I do not want to appear pedantic, but using a correct terminology may be important to make the paper appreciated also by statisticians.

So, if I understood correctly, I suggest to replace "test statistics" with "p-value", or explaining well that in this paper what we mean with the term test statistics. To be clear, in my previous review I have just highlighted the fact the the p-value is not the statistical significance. Here this problem has been solved, because the p-value is used as a quantity to be compared considering eruption periods and quiet phases.

The reviewer understood well. We are calculating and comparing p-values. We agree with the reviewer that the labelling p-value is more adequate and modified it accordingly.

Reviewer #2 (Remarks to the Author):

Thank you for your very detailed comments and responses to my suggestions.

I think you have done a really excellent and thorough job with the changes, and you have pretty much addressed all the main concerns/questions that I had, as well as those of the other reviewers I think, since we had some overlap in our comments - especially in how the results are/can be generalised and transferred to other volcanoes. I think the additional section on transferability has definitely strengthened the overall arguments presented.

Also, I think the changes to the methods section in particular really help, and I am now much clearer on some of the details that were missing or unclear previously, such as the DSAR median and rate variance and outlier removal. The "quick recipes" in the supplementary material also helps clarify the exact processing steps required in order to reproduce the results.

I am happy to recommend and endorse this manuscript for publication

We are grateful to the reviewer for the extensive review provided.

Reviewer #3 (Remarks to the Author):

This is a follow up review after revisions to the manuscript requested in earlier review.

I'm happy that the intent of my comments have been reflected in revisions carried out by the authors. I particularly appreciate the addition of material that better explains how key features were calculated, including quick recipe supplemental section.

line 25/26 the authors noted they'd removed 'sudden' in describing the 2019 Whakaari eruption, but it remains in the text. Maybe it was here twice and only one has been removed???

Modified.

line 42 and related. The discussion on tremor and sealing is a lot better and I think adds substantially to the logic behind how the authors are interpreting the changes in tremor and derived features.

I would be happy to see the manuscript published in it's current form.

We are grateful to the reviewer for the extensive review provided.